# Agl24 is an ancient archaeal homolog of the eukaryotic N-glycan chitobiose synthesis enzymes

Benjamin H Meyer[1,2,3]*, Panagiotis S Adam[4], Ben A Wagstaff[2], George E Kolyfetis[5], Alexander J Probst[6], Sonja V Albers[3], Helge C Dorfmueller[2]*

[1]Environmental Microbiology and Biotechnology (EMB), Aquatic Microbial Ecology, University of Duisburg-Essen, Essen, Germany; [2]Division of Molecular Microbiology, School of Life Sciences, University of Dundee, Dundee, United Kingdom; [3]Molecular Biology of Archaea, Faculty of Biology, University of Freiburg, Freiburg, Germany; [4]Group for Aquatic Microbial Ecology, Environmental Microbiology and Biotechnology, Faculty of Chemistry University Duisburg-Essen, Essen, Germany; [5]Department of Biochemistry and Molecular Biology, Faculty of Biology, National and Kapodistrian University of Athens, Athens, Greece; [6]Centre of Water and Environmental Research (ZWU), University of Duisburg-Essen, Essen, Germany

*For correspondence:
benjamin.meyer@uni-due.de
(BHM);
hczdorfmueller@dundee.ac.uk
(HCD)

**Competing interest:** The authors declare that no competing interests exist.

**Abstract** Protein N-glycosylation is a post-translational modification found in organisms of all domains of life. The crenarchaeal N-glycosylation begins with the synthesis of a lipid-linked chitobiose core structure, identical to that in Eukaryotes, although the enzyme catalyzing this reaction remains unknown. Here, we report the identification of a thermostable archaeal β-1,4-*N*-acetylglucosaminyltransferase, named a̲rchaeal g̲lycosylation enzyme 24 (Agl24), responsible for the synthesis of the N-glycan chitobiose core. Biochemical characterization confirmed its function as an inverting β-D-GlcNAc-(1→4)-α-D-GlcNAc-diphosphodolichol glycosyltransferase. Substitution of a conserved histidine residue, found also in the eukaryotic and bacterial homologs, demonstrated its functional importance for Agl24. Furthermore, bioinformatics and structural modeling revealed similarities of Agl24 to the eukaryotic Alg14/13 and a distant relation to the bacterial MurG, which are catalyzing the same or a similar reaction, respectively. Phylogenetic analysis of Alg14/13 homologs indicates that they are ancient in Eukaryotes, either as a lateral transfer or inherited through eukaryogenesis.

## Editor's evaluation

In this manuscript, Meyer and co-workers further contribute to our understanding of how proteins are modified with carbohydrates on asparagine residues, N-glycosylation. They analyze this in the thermoacidophilic archaea Sulfolobus acidocaldarius and characterize Agl24 as a key player in the N-glycosylation pathway in this organism. These findings advance the glycobiology field and reveal how distinct mechanisms have arisen across evolution to post-translationally modify protein structures.

## Introduction

Asparagine (N)-linked glycosylation is one of the most common co- and post-translational protein modifications found in all three domains of life (*Nothaft and Szymanski, 2010*; *Larkin and Imperiali, 2011*; *Jarrell et al., 2014*). In Eukaryotes, N-glycosylation is evolutionarily highly conserved

(*Lehle et al., 2006*) and it is estimated that more than half of all eukaryotic proteins are glycoproteins (*Apweiler et al., 1999*; *Zielinska et al., 2010*). Correct folding and stability of proteins, intra- and extracellular recognition of protein targets and enzymatic activity necessitate N-glycosylation (*Varki, 1993*; *Helenius and Aebi, 2004*; *Caramelo and Parodi, 2007*). Consequently, protein glycosylation is essential for many cellular processes of Eukaryotes. As such, congenital disorders of glycosylation in humans are often lethal or cause severe phenotypes and diseases, including intellectual disability, growth retardation, cardiac anomalies, and early death (*Reily et al., 2019*). Although it was long thought that N-glycosylation was restricted to Eukaryotes, N-glycosylation pathways have now been characterized in organisms of all domains of life.

The bacterial N-glycosylation processes, which assemble the N-glycan as an intermediated step on a lipid carrier and use an oligosaccharyltransferase (OST) for the transfer of the N-glycan to the target protein, as is has been described for the eukaryotic N-glycosylation, seem to be restricted to Delta- and Epsilon-Proteobacteria. However, also non-lipid-dependent N-glycosylation processes, in which a single nucleotide-activated sugar is directly transferred to a target protein, has also been described in different Bacteria, for example, for *Haemophilus influenza* (*Grass et al., 2010*; *Whitfield et al., 2017*; *Nothaft and Szymanski, 2019*).

By contrast, all sequenced Archaea possess the N-glycan OST AglB, with the exceptions of *Aeropyrum pernix* and *Methanopyrus kandleri* (*Jarrell et al., 2014*; *Nikolayev et al., 2020*). The common feature of N-glycosylation is the assembly of an oligosaccharide onto a lipid through the activity of specific glycosyltransferases (GTs), which use either nucleotide- or lipid-activated sugar donors. The assembled lipid-linked oligosaccharide (LLO) is flipped across the cytoplasmic/ER membrane. The glycan is then either enlarged or directly transferred onto a target protein by an OST. Archaeal N-glycosylation displays a mosaic of bacterial and eukaryotic features. Like Bacteria, Archaea use only a single OST subunit, whereas the majority of Eukaryotes require a large complex of distinct OST subunits to transfer the LLO onto the target protein (*Wild et al., 2018*). The level of complexity of the eukaryotic OST varies across kingdoms; vertebrates require an OST complex of eight subunits, plants and insects of seven, while kinetoplastids, like *Trypanosoma brucei* or *Leishmania major*, possess only a single subunit (*Kelleher and Gilmore, 2006*).

In contrast to Bacteria, which use undecaprenol as the N-glycan lipid carrier, Archaea assemble the N-glycan on dolichol (Dol) identical to Eukaryotes. Archaea also display unique N-glycosylation features, for example, the N-glycan assembly onto Dol-phosphate (Dol-P) in Euryarchaeota, while Crenarchaeota use Dol-pyrophosphate (Dol-PP) as do Eukaryotes (*Taguchi et al., 2016*).

Currently, more than 15 different archaeal N-glycan structures have been solved (*Jarrell et al., 2014*; *Albers et al., 2017*). These archaeal N-glycans display unique structures and sugar compositions. Based on this limited characterization of archaeal N-glycans, it is plausible to assume that Archaea possess an enormous range of structural and composition diverse N-glycans. However, closely related species, for example, in the thermophilic crenarchaeal Sulfolobales family, retain a conserved N-glycan core structure but are differentiated by their terminal sugar residues (*Zähringer et al., 2000*; *Palmieri et al., 2013*; *van Wolferen et al., 2020*). The protein N-glycosylation pathway of *Sulfolobus acidocaldarius* (order: Sulfolobales) has only been partially characterized. Secreted proteins, such as the cell wall surface-layer (S-layer) protein and motility filament forming protein ArlB, are N-glycosylated with a tribranched hexasaccharide. This N-glycan has the composition: (Glc-β1–4-sulfoquinovoseβ-1–3)(Man α-1–6)(Man α-1–4)GlcNAc-β1–4-GlcNAc-β-Asp (*Zähringer et al., 2000*; *Peyfoon et al., 2010*). Interestingly, the N-glycan core structure is composed of a chitobiose (GlcNAc-β1–4-GlcNAc) identical to the core structure of all eukaryotic N-glycans. Members of the Sulfolobales family are the only Archaea reported to display this eukaryotic N-linked chitobiose core structure (*Palmieri et al., 2013*; *van Wolferen et al., 2020*). Recently, the characterization of the N-glycan from *Pyrobaculum calidifontis*, belonging to the Thermoproteales order of the Crenarchaeota, revealed as similar N-glycan chitobiose core structure, with the difference that both GlcNAc are modified with an additional *N*-acetyl group at the C3 position while the second GlcNAc residue is oxidized to glucoronate (*Fujinami et al., 2017*).

Based on this eukaryotic N-glycan core structure, it has been suggested that the *Sulfolobus* N-glycan biosynthesis pathway may be similar to the eukaryotic process (*Meyer and Albers, 2013*). Indeed, a homolog of the yeast Alg7 enzyme, which initiates eukaryotic N-glycosylation by transferring a GlcNAc-P from UDP-GlcNAc to Dol-P to generate the precursor Dol-PP-GlcNAc, has been identified

in *S. acidocaldarius* as AlgH (*Meyer et al., 2017*). Not only did this AglH show a similar topology and highly conserved amino acid motifs to Alg7, but also functionally complemented a conditional yeast Alg7 knockout (*Meyer et al., 2017*). This functional complementation has been first described for the AglH of *Methanococcus voltae* (*Shams-Eldin et al., 2008*), which has suggested that AglH initiates the N-glycosylation by transferring GlcNAc-1-P onto Dol-P. However, a biochemical study clearly showed that AglK is responsible for the generation of Dol-P-GlcNAc, on which the N-glycan is assembled (*Larkin et al., 2013*). This observation agrees that Euryarchaeota build their N-glycans on Dol-P and not on Dol-PP in a similar manner to Crenarchaeaota and Eukaryotes (*Taguchi et al., 2016*).

Here, we report the identification of the β-1,4-*N*-acetylglucosaminetransferase Agl24 from *S. acidocaldarius*, which shows conserved amino acid motifs and apparent structural similarities with the eukaryotic Alg14/13 and bacterial MurG functional homologs. Activity assays along with HPLC, matrix-assisted laser desorption/ionization (MALDI), and NMR analyses confirmed the function of Agl24 as catalyzing the second N-glycan biosynthesis step to generate the lipid-linked chitobiose core. Our extensive bioinformatics analyses support the hypothesis that the eukaryotic N-glycosylation pathway originates from an archaeal ancestor.

## Results

### Identification of a Dol-PP-GlcNAc UDP-GlcNAc GlcNAc GT candidate in *S. acidocaldarius*

To identify the GT that catalyzes chitobiose biosynthesis step in the N-glycosylation process in *S. acidocaldarius*, we searched for similar biosynthetic processes to identify potential homologs. The bacterial enzyme MurG transfers a GlcNAc residue from UDP-GlcNAc to the C4 hydroxyl group of MurNAc-(pentapeptide)-PP-undecaprenol to produce the lipid-linked β-(1,4) disaccharide known as lipid II (*Men et al., 1998*; *Ha et al., 1999*). Although MurG does not synthesize chitobiose-like *S. acidocaldarius* does, the enzyme uses a similar lipid-linked acceptor and an identical sugar donor (UDP-GlcNAc) and therefore presents a valid candidate for evaluation. MurG has been classified in the Carbohydrate Active Enzyme database (http://www.cazy.org/) as a family 28 GT (*Lombard et al., 2014*). Until now, only a few archaeal organisms possess GT28 family homologs. These organisms include members of the archaeal orders Methanobacteriales and Methanopyrales, which have been described that synthesize pseudomurein as a component of their cell wall structure (*Kandler and König, 1978*; *König et al., 1982*; *Albers and Meyer, 2011*), and *Salinadaptatus halalkaliphilus* 2447, a recently described member of the Halobacteriales.

Another candidate homolog is the eukaryotic protein complex formed by Alg13 and Alg14 that transfers GlcNAc from UDP-GlcNAc to the C4 hydroxyl of GlcNAc-PP-Dol to produce the β-(1-4) chitobiose core of the eukaryotic N-glycan. Alg13 and Alg14 are classified as GT1 family members but are distantly related to GT28 family enzymes (*Lombard et al., 2014*). Alg13 and Alg14 form the C- and N-terminal domains, respectively, of a functional heterodimeric GT enzyme; Alg14 contains the acceptor-binding site and the membrane association domain, while Alg13 possesses the UDP-GlcNAc-binding pocket. The interaction of these two enzymes is required for the catalytic activity, as the active site lies in the cavity formed by the two proteins (*Bickel et al., 2005*; *Chantret et al., 2005*; *Gao et al., 2005*). Although members of the GT1 family are prominent in Archaea, until now no GT1 family member has been identified in genomes of Sulfolobales.

As Alg14-13 and MurG use UDP-GlcNAc to GlcNAcylate a lipid-linked sugar acceptor, they represent valid candidates for a bioinformatics search in *S. acidocaldarius*.

Using the *Escherichia coli* MurG protein sequence (WP_063074721.1; EC 2.4.1.227) as template for a Delta BLAST search (*Boratyn et al., 2012*), various homologs were identified in *S. acidocaldarius* with very low sequence identity of only 10–17% (*Supplementary file 1*). The anchoring subunit Alg14 from *Saccharomyces cerevisiae* S288C (NP_009626.1) was used as query for a Delta BLAST homology search but no protein candidate was identified below the e-value threshold of 0.005. However, when either the catalytic subunit Alg13 from *S. cerevisiae* S288C (NP_011468.1; EC 2.4.1.141) or the fusion of both Alg14 and Alg13 sequences was used as a search model, a single Sulfolobales homolog was identified with 17% sequence identity (*Supplementary file 1*). In addition, phylogenetic studies indicated similarities of MurG and Alg14/13 with Saci1262 (*Lombard, 2016*; *Cavalier-Smith and Chao,*

*2020*). Thus, the most likely candidate for the β-1,4-*N*-acetylglucosaminetransferase in Sulfobales was found to be Saci1262 (Uniprot: Q4J9C3), a hypothetical 327 amino acid long protein.

In contrast to Euryarchaeota, where the genes of the N-glycosylation enzymes are clustered with the OST *aglB,* clustering of GT genes is uncommon in Crenarchaeota, including Sulfolobales (*Kaminski et al., 2013*; *Nikolayev et al., 2020*). Interestingly, the gene *saci1262* is located only eight genes downstream of the OST *aglB* (*saci1274*) in *S. acidocaldarius* (*Figure 1—figure supplement 1*), while the genes coding for identified enzymes involved in the N-glycosylation, for example, *aglH* (*saci0093*), *agl3* (*saci423*), and *agl16* (*saci0807*) are distantly scattered across the entire genome.

## Detailed bioinformatics comparison of Saci1262 with Alg14/13 and MurG

Sequence and domain comparisons of the eukaryotic Alg13 and Alg14 to MurG and Saci1262 revealed that Alg14 corresponds to the N-terminal part of MurG/Saci1262, while Alg13 resembled the C-terminal part of MurG/Saci1262 (*Figure 1A*). Both the N-terminal (aa1–152) and C-terminal (aa153–327) parts of Saci1262 showed very low sequence identity to Alg14 and Alg13, with 15% and 16%, respectively. Interestingly, fused homologs of Alg14 and Alg13 have been reported in the Kinetoplastea *Leishmania* and *Trypanosoma*, similar to *S. acidocaldarius* (*Averbeck et al., 2007*). While a fusion with the reverse domain order was recognized in the Eukaryotes *Entamoeba histolytica* (*Bickel et al., 2005*) and *Dictyostelium discoideum* (*Averbeck et al., 2007*; *Figure 1A*). An interesting difference in these proteins from the three domains of life is the presence of an N-terminal transmembrane domain (TMD) in most of eukaryotic Alg14 proteins (aa 5–25), which is missing in the bacterial MurG, archaeal Saci1262-like enzymes, as well as in the Alg13-14 fusion of *E. histolytica* and *D. discoideum*. The TMD is predicted to facilitate the association of Alg14 with the ER membrane and to participate in recruitment of Dol-PP-GlcNAc to the membrane (*Lu et al., 2012*). The introduction of this TMD appears to have emerged later in the evolution of Eukaryotes, and the interaction with the ER or cytoplasmic membrane is not solely dependent on a TMD, as the presence of hydrophobic patches and protein–protein interactions with membrane proteins are also proposed to facilitate this interaction (*Ha et al., 2000*).

The amino acid sequence alignment of the representative bacterial MurG, eukaryotic Alg14-13, and archaeal homologs revealed conservation of specific patches and individual residues (*Figure 1B and D*). The overall topology of Saci1262 is identical to MurG, which consists of two domain folds separated by a deep cleft corresponding to a GT-B fold structure (*Ha et al., 2000*; *Hu et al., 2003*; *Figure 1C*). Despite the low sequence identity, a similar structural topology of Saci1262 to Alg14-13 and MurG was not surprising, as GTs in general showed only a limited number of structural folds. GTs mainly possessed a GT-A or GT-B fold, whereas OSTs and a number of recently characterized GTs displayed a GT-C, GT-D, or GT-E fold, respectively (*Mestrom et al., 2019*). Crystal structures of the bacterial MurG (PDB:3s2U, 1F0K, 1NLM) and the C-terminal half of the eukaryotic Alg13 (PDB:2jzc) showed typical GT-B structural characteristics (*Ha et al., 2000*; *Hu et al., 2003*; *Wang et al., 2008*; *Raman et al., 2010*; *Brown et al., 2013*; *Figure 2*). The N-terminal part of Alg13 contains a Rossmann-like fold with a mixed parallel and antiparallel β-sheet rather than the conventional Rossmann fold found in all GT-B enzymes, indicating a unique topology among GTs (*Wang et al., 2008*). The N-terminal helix of Alg14 and MurG is proposed to be the membrane association domain (*Chantret et al., 2005*; *Lu et al., 2012*). We have generated a Saci1262 structural model to investigate potential features via SWISS-MODEL (*Waterhouse et al., 2018*), which is in agreement with the model obtained from AlphaFold (*Jumper et al., 2021*; *Figure 2*, doi.org/10.7910/DVN/9KSWQR). Previous studies have compared the structures of MurG and Alg13, including a model for Alg14 (*Gao et al., 2008*). The formation of the Alg14-Alg13 complex is mediated by a short C-terminal α-helix of Alg13 in cooperation with the last three amino acids of Alg14 (*Gao et al., 2008*). In the Alg13 crystal structure, this C-terminal helix associates with itself, as Alg14 is missing. The orientation is predicted to be similar to the linker in MurG and the change is indicated by the black arrow (in *Figure 2A*; *Gao et al., 2008*). The structural model of Saci1262 is also predicted to contain a C-terminal helix, which interacts with the N-terminal part of the protein *Figure 2A*.

We discovered a conserved GGxGGH$_{14}$ motif (Saci1262) within the N-terminal sequences of MurG, Alg14, and Saci1262-like proteins (*Figure 1D*, *Figure 1—figure supplement 2*, and *Figure 2B*). This motif has been described in MurG sequences in an extended form (G$^{13}$GTGGHX$_2$PXLAXAX$_2$LX$_9$G$^{39}$)

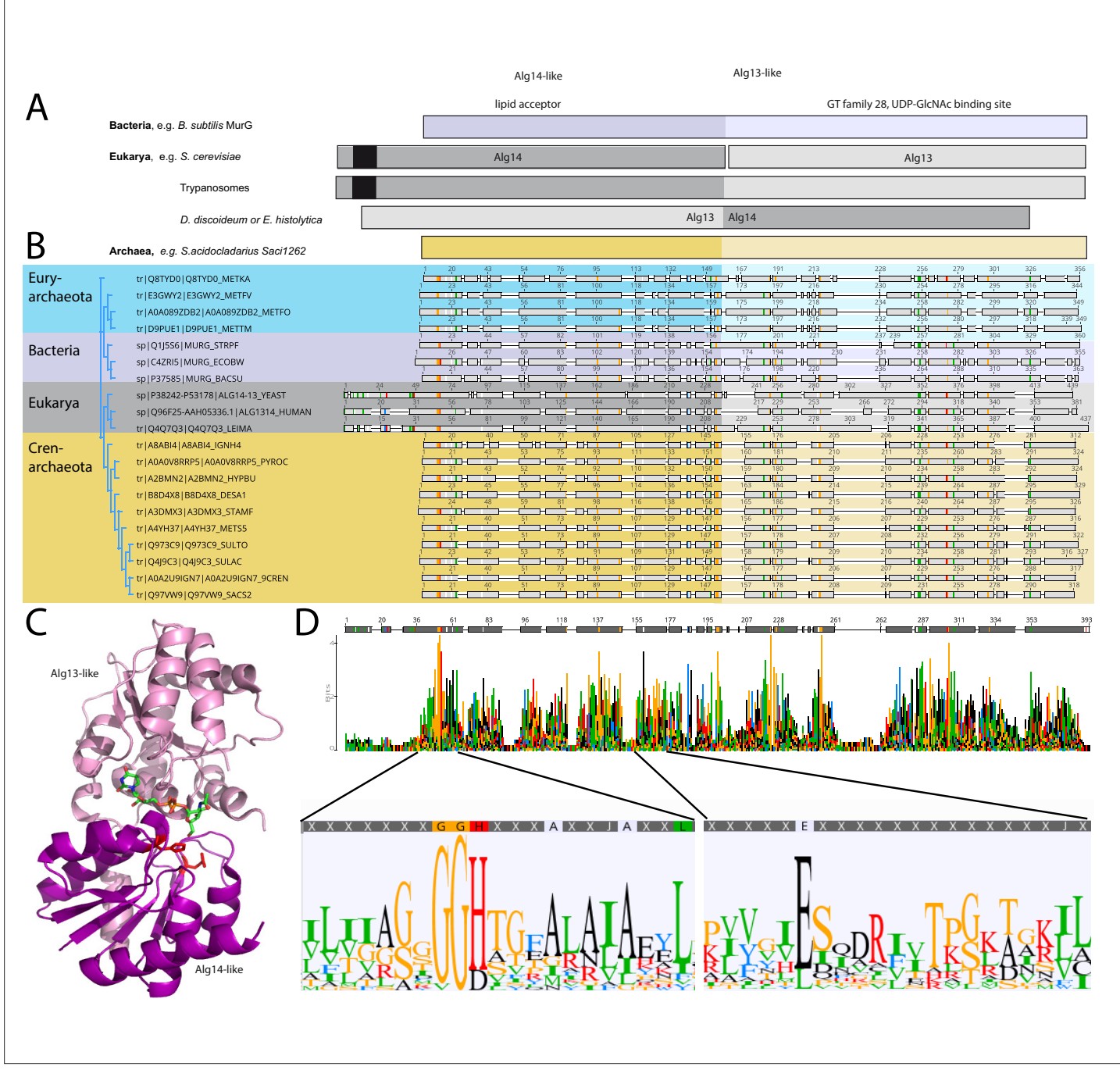

**Figure 1.** Comparison of the archaeal Saci1262, bacterial MurG, and eukaryotic Alg14-13. (**A**) Simplified comparison of the Alg14-like (darker) or Alg13-like (light color) domain in bacterial MurG (violet), eukaryotic Alg14-13 (gray), and archaeal Saci1262 (orange) homologs. The transmembrane domain is depicted in black. (**B**) Protein sequences were aligned with Clustal Omega (**Sievers and Higgins, 2018**) (for the full alignment, see **Figure 1—figure supplement 2**). Sequences derived from (i) three bacterial (violet background) MurGs: *Escherichia coli* (C4ZRI5), *Streptococcus pyogenes* (Q1J5S6), and *Bacillus subtilis* (P37585); (ii) eukaryotic (gray background) Alg14-13: *Leishmania major* (Q4Q7Q3), an artificial Alg14-13 fusion from *Saccharomyces cerevisiae* (P38242-P53178) and *Homo sapiens* (Q96F25-Q9NP73-2); (iii) the crenarchaeal (orange background) Saci1264 homologs: *Ignicoccus hospitalis* (A8ABI4), *Pyrodictium occultum* (A0A0V8RRP5), *Hyperthermus butylicus* (A2BMN2), *Desulfurococcus amylolyticus* (B8D4 × 8), *Staphylothermus marinus* (A3DMX3), *Metallosphaera sedula* (A4YH37), *Sulfurisphaera tokodaii* (Q973C9), *Sulfolobus acidocaldarius* (Q4J9C3), *Acidianus brierleyi* (A0A2U9IGN7), and *Saccharolobus solfataricus* (Q97VW9). Selected sequences from pseudomurein producing Euryarchaeota with higher sequence similarity to the MurG are aligned (turquoise background): *Methanopyrus kandleri* (Q8TYD0), *Methanothermus fervidus* (E3GWY2), *Methanobacterium formicicum* (A0A089ZDB2), *Methanothermobacter marburgensis* (D9PUE1). Conserved amino acids (65% threshold) are highlighted in color corresponding to the amino acids: G, S, T, P (orange), K, R, H (red), F, W,Y (blue), A, D, E, C, Q, N (black), and V, L, I, M (green). The end of the Alg14-like domain and start of

*Figure 1 continued on next page*

*Figure 1 continued*

Alg13-like domains are indicated by the change from dark to light background color. (**C**) MurG (PDB: 3s2u; shown as ribbon) bound to UDP-GlcNAc (shown as sticks, green carbon, blue nitrogen, red oxygen) and orientation toward the cell membrane (bottom). The Alg13-like domain (light violet) and Alg14-like domain (dark violet) are labelled. Conserved His and Glu residues in Alg14 are shown as red sticks. (**D**) Strict consensus (65%) and Weblogo from the sequence alignment shown in (B). Bar heights correspond to the observed frequency, highly conserved motifs or amino acids are enlarged below, including conserved motifs G(x)GGH$_{15}$ (Loop I) and E$_{114}$ (for the full Weblogo, see **Figure 1—figure supplement 2**).

The online version of this article includes the following figure supplement(s) for figure 1:

**Figure supplement 1.** Physical map of the gene region adjacent to *saci1262 and aglB* of *Sulfolobus acidocaldarius*.

**Figure supplement 2.** Protein sequences were aligned with Clustal Omega (**Sievers and Higgins, 2018**).

(**Crouvoisier et al., 2007**). The conserved amino acids are located in the cavity between the two different structural domains next to the substrate-binding pocket. The crystal structure of MurG (PDB: 3s2u) in complex with UDP-GlcNAc shows the close proximity of the sugar donor to the conserved residues (**Figure 2B**; **Brown et al., 2013**). In proximity to the GGxGGH$_{14}$ motif is the conserved glutamic acid (E$_{114}$) (**Figure 1D**, **Figure 1—figure supplement 2**, and **Figure 2B**). The two conserved residues H$_{14}$ and E$_{114}$ are absent in the archaeal MurG-like GT28 family homologs of known pseudomurein-producing Archaea. Instead, aspartic acid (D) and leucine (L) are more frequently found replacing the H$_{14}$ and E$_{114}$ (**Figure 1D** and **Figure 1—figure supplement 2**). We propose that these residues might be required to accommodate the different sugar *N*-acetyltalosaminuronic acid of the lipid-linked acceptor molecule in the pseudomurein biosynthesis process.

## The *Saci1262* gene is essential in *S. acidocaldarius*

Due to the overall low sequence similarity of Saci1262 to Alg14-13 and MurG, we attempted to demonstrate the predicted function of Saci1262 in vivo by generating a deletion mutant of *saci1262* in the genome of *S. acidocaldarius*. This deletion mutant should arrest the N-glycosylation process after the generation of Dol-PP-GlcNAc and should contain either non-glycosylated glycoproteins or N-glycosylated proteins containing only a single GlcNAc residue linked to the asparagine in the

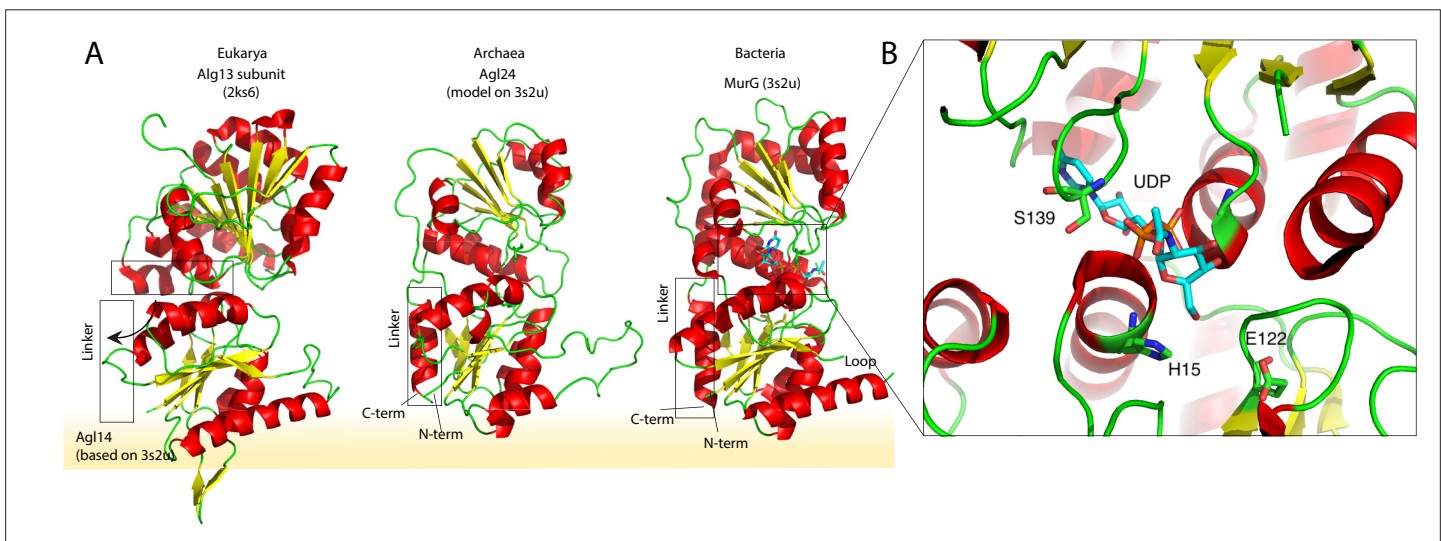

**Figure 2.** Structural comparison of the eukaryotic and bacterial homologs of the archaeal Saci1262. (**A**) Structural comparison of the eukaryotic Alg13 (PDB:2ks6) with a structural model of Alg14, the bacterial MurG (PDB: 3s2u), and the structural model of the crenarchaeal Saci1262 (archaeal glycosylation enzyme 24 [Agl24]). Structural models were built with SWISS-MODEL (**Waterhouse et al., 2018**). Detailed results are listed in **Supplementary file 2**. The interactions of the N-terminal helices with the membrane are depicted with a yellow background. (**B**) Magnified and 90° rotated view into the active site of MurG (PDB: 3s2u) in complex with UDP-GlcNAc (sticks). The catalytic site is located in the cleft of both domains, with conserved H15 and E122 residues shown in sticks with green carbon, red oxygen, and blue nitrogen atoms.

The online version of this article includes the following figure supplement(s) for figure 2:

**Figure supplement 1.** Confirmation of the integration and segregation of the *saci1262* deletion plasmid pSVA1312 in *Sulfolobus acidocaldarius* MW001.

conserved N-glycosylation motif. The genomic integration by homologous recombination of the plasmid pSVA1312 via either the up- or downstream region of the *saci1262* and the selection *pyrEF* genes was confirmed by PCR (*Figure 2—figure supplement 1A*). However, we were not successful in generating the marker-less in-frame *saci1262* deletion mutant. An alternative approach was conducted aiming at deleting the *saci1262* gene by a single homologous recombination step by the integration of the *pyrEF* selection cassette. All attempts to generate this gene disruption mutant failed. This strongly suggests that the *saci1262* gene is essential in *S. acidocaldarius*, at least under the conditions tested. To exclude that the other GTs, identified in the bioinformatic homology search (Table S1), are involved in the N-glycosylation, marker-less deletion mutants of *saci1907*, *saci1921*, *saci0807*, *saci1094*, *saci1201*, *saci1821*, and *Saci1249* were successfully generated. Only the deletion of *saci0807*, which has been characterized to encode the GT Agl16 that transfers the terminal glucose residue to the N-glycan (*Meyer et al., 2013*), showed a significant effect on the N-glycosylation of the S-layer proteins. Since we have previously shown that N-glycosylation is essential in Sulfolobus, and saci1262 the only candidate gene is which is essential, our data suggested that the protein encoded by *saci1262* is potentially the second enzyme synthesizing the N-glycosylation chitobiose core.

## Saci1262 transfers a single GlcNAc residue onto a GlcNAc pyrophosphate-linked acceptor molecule

In agreement with the observed membrane association in *S. acidocaldarius* (*Figure 3—figure supplement 1*), Saci1262 was also found in the membrane fraction during its purification from *E.*

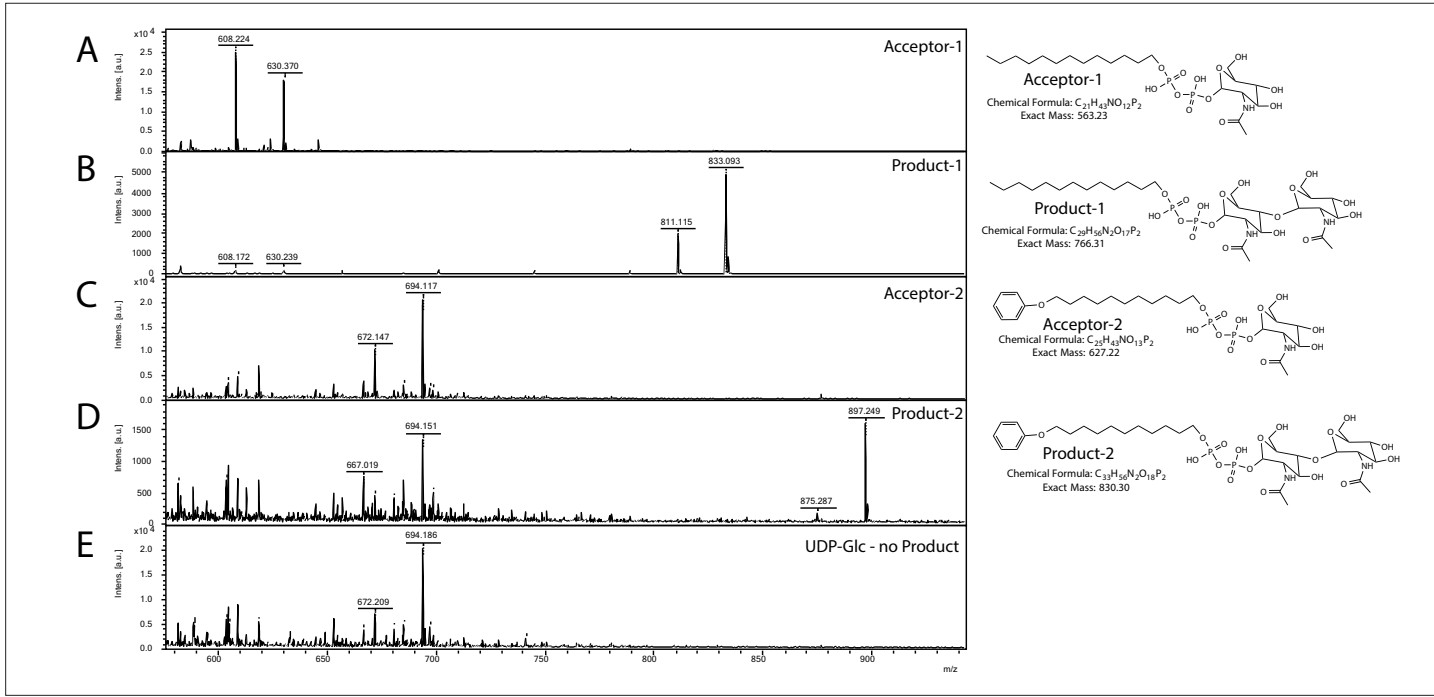

**Figure 3.** Matrix-assisted laser desorption/ionization mass spectrometry (MALDI-MS) spectra of the in vitro Saci1262 reaction assessing the *N*-acetylglucosaminetransferase activity. Spectra obtained from the enzymatic reactions containing Saci1262-GFP and (**A**) acceptor-1, (**B**) acceptor-1 and UDP-GlcNAc, and Saci1262-GFP, (**C**) acceptor-2, (**D**) acceptor-2, and UDP-GlcNAc, (**E**) acceptor-2 and UDP-Glc. The conversion from acceptor-1 (608 m/z [M-1H + 2Na]$^+$ and 630 m/z [M-2H + 3Na]$^+$) or acceptor-2 (672 m/z [M-1H + 2Na]$^+$ and 694 m/z [M-2H + 3Na]$^+$) to the product (811 m/z [M-1H + 2Na]$^+$ and 833 m/z [M-2H + 3Na]$^+$) or (875 m/z [M-1H + 2Na]$^+$ and 897 m/z [M-2H + 3Na]$^+$) was observed only when UDP-GlcNAc was used as nucleotide sugar donor.

The online version of this article includes the following figure supplement(s) for figure 3:

**Figure supplement 1.** Anti-His Immunoblot indicates membrane association of Saci1262, while Alg3 (sulfoquinovose synthase) is only found in the soluble fraction.

**Figure supplement 2.** SDS-PAGE of the purified Agl24-WT-GFP, mutants Agl24-H15A-GFP, and Agl24-E114A-GFP from *Escherichia coli*, stained with Coomassie Brilliant Blue or visualized by in-gel fluorescence (IGF).

**Figure supplement 3.** HPLC chromatogram and kinetic parameters of the archaeal glycosylation enzyme 24 (Agl24) enzymatic reaction.

coli. Recombinant Saci1262, produced in *E. coli* (*Figure 3—figure supplement 2*), was assayed to test the function of Saci1262 using the predicted nucleotide sugar donor UDP-GlcNAc and two synthetic acceptor substrates designed to mimic the native lipid-linked acceptor: $C_{13}H_{27}$-PP-GlcNAc (acceptor-1) or phenyl-O-$C_{11}H_{22}$-PP-GlcNAc (acceptor-2) (*Figure 3A and C*). The reaction products were purified and characterized. The MALDI-MS spectra obtained confirmed that Saci1262 transfers a single GlcNAc to both acceptor substrates when incubated with UDP-GlcNAc (*Figure 3B and D*). The Saci1262 assay revealed a shift of the peaks by 203 Da to m/z = 811 [M-1H + 2Na]$^+$ and m/z = 833 [M-2H + 3Na]$^+$ (*Figure 3B*), which corresponded to the addition of one dehydrated GlcNAc (203 Da). The same shift was observed for acceptor-2 (*Figure 3D*). With an extended reaction time, no additions of a further GlcNAc by Saci1262 could be detected. We also investigated Saci1262 promiscuity toward utilizing UDP-glucose as the sugar donor, but no mass shift was observed (*Figure 3E*), indicating that Saci1262 uses exclusively UDP-GlcNAc. Based on the confirmed function of Saci1262 as a GT, we named the protein as archaeal glycosylation enzyme 24 (Agl24), in agreement with the general naming procedure of N-glycosylation pathway components in Archaea (*Eichler et al., 2013*).

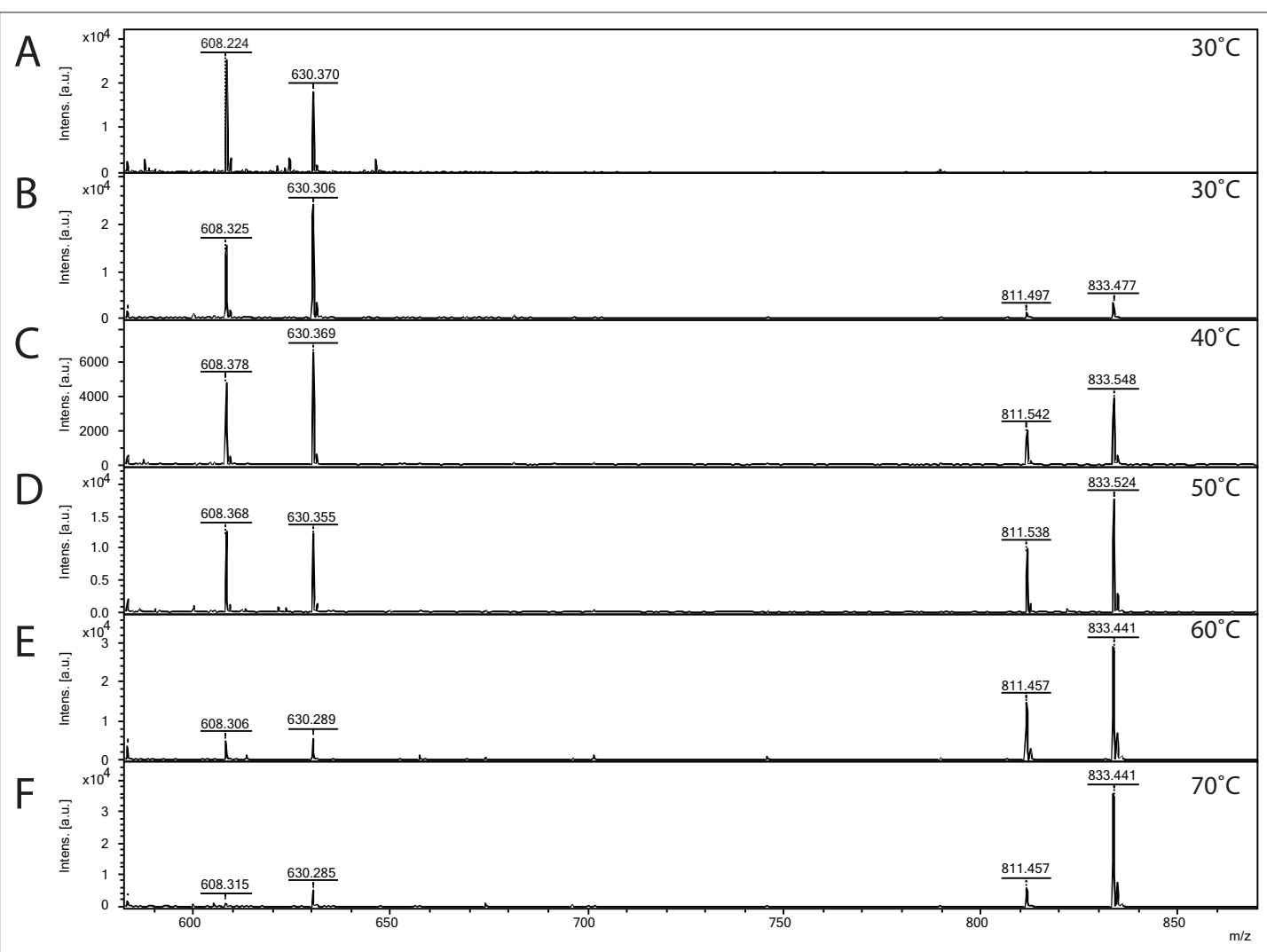

**Figure 4.** Matrix-assisted laser desorption/ionization mass spectrometry (MALDI-MS) spectra of the in vitro archaeal glycosylation enzyme 24 (Agl24) reaction at different temperatures. Spectra obtained from the purified enzymatic reaction mix with only acceptor-1 (**A**), acceptor-1, UDP-GlcNAc, and Agl24-GFP wild-type (WT) at 30°C (**B**), 40°C (**C**), 50°C (**D**), 60°C (**E**), and 70°C (**F**). Conversion of the acceptor-1 (608 m/z [M-1H + 2Na] and 630 m/z [M-2H + 3Na]) toward the product (811 m/z [M-1H + 2Na] and 833 m/z [M-2H + 3Na]) is dependent on the applied temperature. Above 50°C, the increase in the level of the product at the expense of the level of the acceptor is clearly visible.

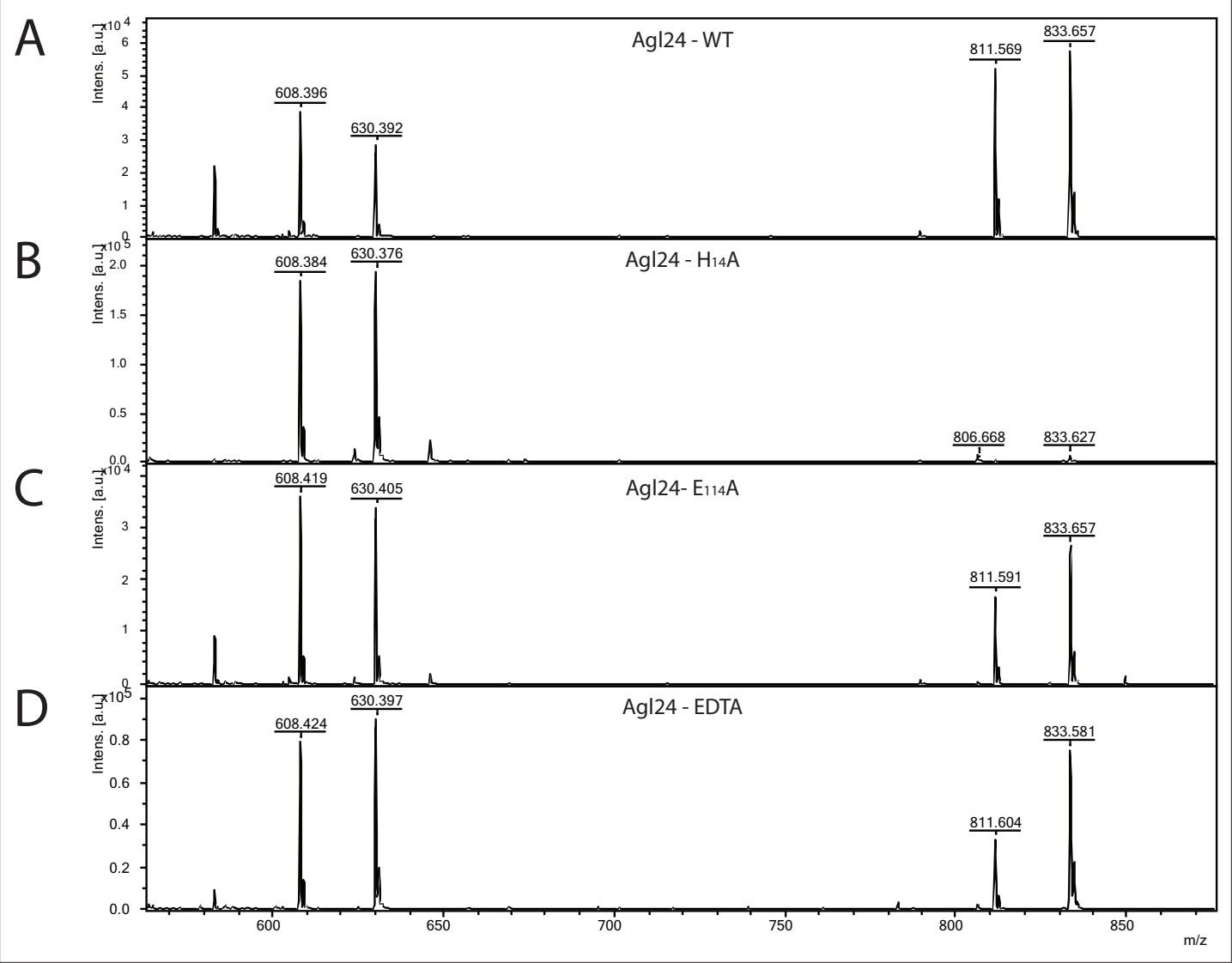

**Figure 5.** Matrix-assisted laser desorption/ionization mass spectrometry (MALDI-MS) spectra of the in vitro archaeal glycosylation enzyme 24 (Agl24) reaction, the generated point mutations H14A and E114A and EDTA control. Spectra obtained from the purified enzymatic reaction mix with acceptor-1 (608 m/z [M-1H + 2Na] and 630 m/z [M-2H + 3Na]), UDP-GlcNAc and (**A**) Agl24-GFP wild-type (WT), (**B**) Agl24-H$_{14}$A-GFP, (**C**) Agl24-E$_{114}$A-GFP, and (**D**) Agl24-GFP WT with addition of 10 mM EDTA. The activity was significantly reduced in the H$_{14}$A mutants, while a similar amount of product (811 m/z [M-1H + 2Na] and 833 m/z [M-2H + 3Na]) was obtained in the E$_{114}$A mutant.

The online version of this article includes the following figure supplement(s) for figure 5:

**Figure supplement 1.** Matrix-assisted laser desorption/ionization mass spectrometry (MALDI-MS) spectra of the in vitro archaeal glycosylation enzyme 24 (Agl24) reaction to assess the metal dependency for the specificity activity.

### Activity of Agl24

Since *S. acidocaldarius* is a thermophilic microorganism, with an optimal growth temperature of 75°C (**Brock et al., 1972**), the temperature dependency of the Agl24 activity was investigated using our established mass spectrometry assay. The substrates are stable at the conditions tested and MALDI analysis of the negative control lacking the enzyme revealed only the acceptor-1 mass (**Figure 4A**). At elevated temperatures, the peak intensity from the product increased while the peak intensity of the acceptor molecule was reduced (**Figure 4B–F**). Highest activity was detected at 70°C, close to the optimal growth temperature of *S. acidocaldarius*. Furthermore, the addition of EDTA did not affect the activity of Agl24, demonstrating that Agl24 is a metal-independent GT (**Figure 5**, **Figure 5— figure supplement 1**). This result agrees with the lack of a conserved Asp-X-Asp (DxD) motif in the

Agl24 sequence, which has been shown to be important to coordinate the metal ions in A-fold GTs, whereas GT-B GTs are metal ion-independent (*Lairson et al., 2008*; *Gloster, 2014*).

## Conserved His14 is essential for Agl24 function

Two conserved amino acid residues were targeted by mutagenesis to investigate their role in the Agl24 enzyme. The product where the $H_{14}$ histidine within the conserved $GGH_{14}$ motif, found in all Alg14 (GSGGH), MurG (GGxGGH), and Agl24 (GGH) homologs, was replaced by an alanine, was inactive (*Figure 5B*). This demonstrated that this highly conserved residue, located next to the nucleotide-binding site, is important for enzyme function. This GGxGGH motif and a subsequent second glycine-rich motif (GGY) have been proposed to enable MurG to be involved in an interaction with the diphosphate group of the lipid acceptor, as these two motifs resemble phosphate-binding loops of nucleotide-binding proteins (*Baker et al., 1992*; *Carugo and Argos, 1997*). Alanine substitution of the conserved His residue in MurG resulted in undetectable activity and loss of ability to complement a temperature-sensitive MurG mutant (*Hu et al., 2003*; *Crouvoisier et al., 2007*). By contrast, the substitution of the conserved residue $E_{114}$, opposite of the nucleotide-binding site, had no effect on the activity of Agl24 (*Figure 5C*). In MurG this glutamic acid residue is found in a conserved $H_{124}EQN_{127}$ motif (*E. coli*), proposed to coordinate the lipid acceptor (*Hu et al., 2003*; *Crouvoisier et al., 2007*). The MurG $E_{125}A$ variant was able to complement the thermosensitive MurG *E. coli* strain, but the remaining activity was 860-fold lower than that of the wild-type MurG (*Crouvoisier et al., 2007*).

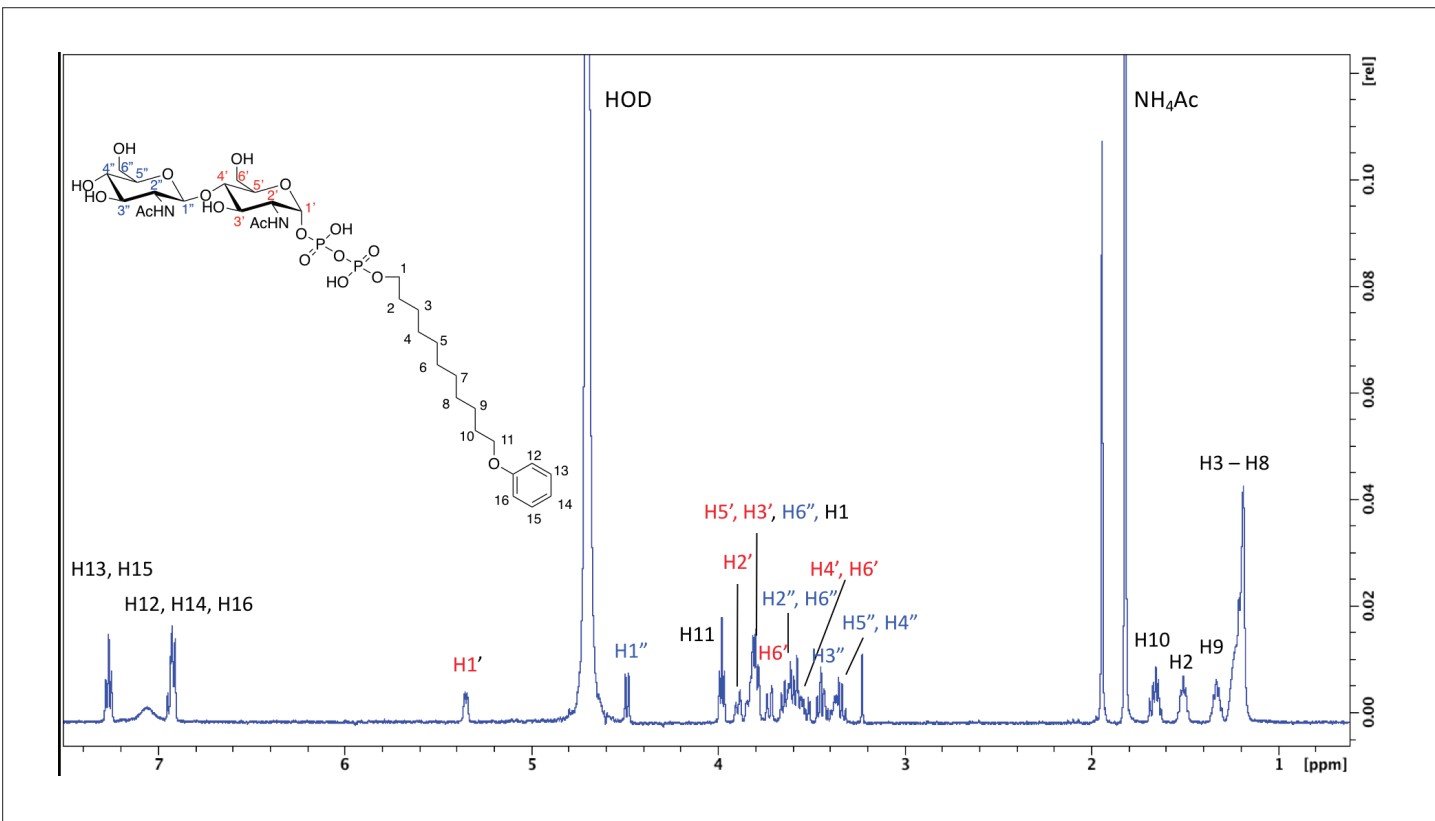

**Figure 6.** $^1$H NMR spectra of purified archaeal glycosylation enzyme 24 (Agl24) enzymatic product. Spectra were acquired in $D_2O$ at 293 K on a Bruker AVANCE III HD 500 MHz NMR spectrometer fitted with a 5 mm QCPI cryoprobe. Chemical shifts are reported with respect to the residual HDO signal at $\delta_H$ 4.70 ppm.

The online version of this article includes the following figure supplement(s) for figure 6:

**Figure supplement 1.** 1D total correlation spectroscopy (TOCSY) correlation spectra.

**Figure supplement 2.** 2D COSY spectra used to assign proton signals arising from α-GlcNAc (top) and β-GlcNAc (bottom).

**Figure supplement 3.** 2D HSQC spectra used to assign carbon signals arising from both α- and β-linked GlcNAc residues.

**Table 1.** $^1$H and $^{13}$C chemical shifts of the in vitro archaeal glycosylation enzyme 24 (Agl24) reaction product.

| Lipid | | 1 | 2 | **3–8** | 9 | 10 | 11 | 12, 14 | 13, 15 | 16 |
|---|---|---|---|---|---|---|---|---|---|---|
| | $^1$H NMR | 3.79 | 1.51 | 1.16–1.26 | 1.33 | 1.66 | 3.98 | 6.92 | 7.26 | 6.92 |
| | $^{13}$C NMR | 66.8 | 30.3 | 28.8 | 28.7 | 28.7 | 68.8 | 115.6 | 130.1 | 121.8 |
| α-GlcNAc | | 1′ | 2′ | 3′ | 4′ | 5′ | 6′a | 6′b | | |
| | $^1$H NMR | 5.35 | 3.896 | 3.8 | 3.58 | 3.83 | 3.73 | 3.57 | | |
| | $^{13}$C NMR | 94.25 | 53.65 | 70 | 79.65 | 71.69 | 60.15 | | | |
| β-GlcNAc | | 1″ | 2″ | 3″ | 4″ | 5″ | 6″a | 6″b | | |
| | $^1$H NMR | 4.49 | 3.64 | 3.45 | 3.34 | 3.38 | 3.8 | 3.6 | | |
| | $^{13}$C NMR | 101.55 | 56.14 | 74.01 | 70.38 | 76.27 | 61.15 | | | |

## Agl24 is an inverting β-1,4-*N*-acetylglucosamine transferase

For a kinetic analysis of Agl24, an HPLC assay was used to monitor the conversion of the mono-GlcNAcylated-PP-lipid acceptor-2 substrate to the bi-GlcNAcylated product (*Figure 3—figure supplement 3A, B*). A $K_m$ value for acceptor-2 could not be determined using this system, due to significant substrate inhibition above 0.5 mM, however, a $K_m^{(app)}$ for UDP-GlcNAc was determined: $^{(UDP-GlcNAc)}K_m$ = 1.37 ± 0.13 mM, $V_{max}$ = 32.5 ± 0.9 pmol min$^{-1}$. These results agree with the $K_m$ values of other GT enzymes, which typically exhibit $K_m$ affinities for their respective nucleotide sugars in the high micromolar range, reflecting the estimated intracellular concentrations of the nucleotide sugar (*Varki et al., 1999*). A $^1$H NMR analysis confirmed that the enzymatic product contained two GlcNAc residues with different linkage types based on the presence of two differently coupled anomeric protons (*Figure 6* and *Figure 6—figure supplement 1*). One anomeric proton appeared as a doublet of doublets (5.35 ppm, $J_{H1-H2}$ = 3.1 Hz, $J_{H1-P}$ = 7.2 Hz), typical of an α-linked GlcNAc residue (*Table 1*). Another anomeric signal appeared as a doublet with a large $J$ coupling value (4.49 ppm, $J_{H1-H2}$ = 8.5 Hz), indicative of a β-linked GlcNAc residue. The substrate acceptor-2 contained an α-linked GlcNAc; this strongly suggested the terminal GlcNAc introduced by Agl24 was β-linked. Experiments using 1D total correlation spectroscopy (TOCSY, *Figure 6—figure supplement 1*) and 2D COSY (*Figure 6—figure supplement 2*) deciphered the detailed proton signals from each of the two sugar rings. In addition, HSQC experiments (*Figure 6—figure supplement 3*) were used to assign the identity of each proton and carbon signal. The C4 signal of the α-GlcNAc residue was shifted to 79.6 ppm, strongly suggesting that the terminal β-GlcNAc was linked at this position. This was further supported by a relative increase in the shift of the H4 proton of the α-GlcNAc residue compared to the un-modified acceptor (*Zorzoli et al., 2019*). Relative shifts of other protons reported for the acceptor aligned with our experimental data. In conclusion, a combination of $^1$H NMR and 1D and 2D TOCSY, COSY, and HSQC experiments confirmed that Agl24 is an inverting β-1,4-GlcNAc transferase.

## The eukaryotic GTs Alg14 and Alg13 are closely related to Asgard homologs

In order to determine the phylogenetic relationship between the eukaryotic N-glycosylation GTs Alg13 and Alg14 with their archaeal and bacterial homologs, an extensive phylogenetic analysis was performed. We reconstructed phylogenies of Alg13/EpsF (*Figure 7A*) and Alg14/EpsE (*Figure 7B*) with bacterial MurG sequences as outgroup. Agl24-like homologs form a clade containing Crenarchaeota, Bathyarchaeota, and one Baldrarchaeota (Asgard). These Agl24-like sequences are found in all Crenarchaeota genomes from the orders Fervidicoccales, Acidilobales, Desulfurococcales, and Sulfolobales, but not in members of the order Thermoproteales (*Figure 7—figure supplement 1*, *Figure 7—figure supplement 2*, *Figure 7—figure supplement 5*, and doi.org/10.7910/DVN/9K-SWQR). The Bathyarchaeota (14 sequences, mainly from Genome Taxonomy Database (GTDB) order o__B26-1) form a strongly supported monophyletic branch within Crenarchaeota (*Figure 7—figure supplement 1* and *Figure 7—figure supplement 2*) resulting from a single lateral transfer event.

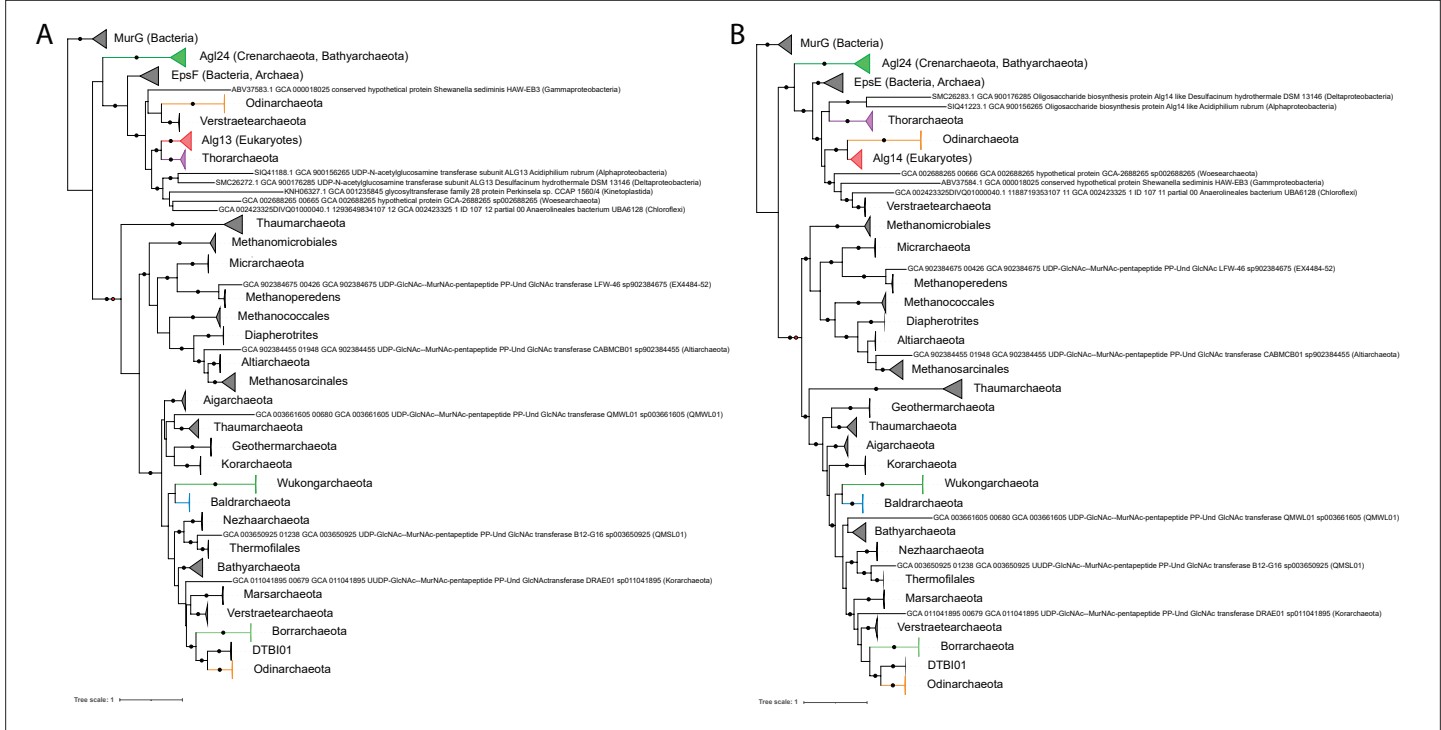

**Figure 7.** Single gene phylogenies of (**A**) Alg13/EpsF and (**B**) Alg14/EpsE homologs. The phylogenies were rooted using the MurG sequences from the corresponding (**Lombard, 2016**) datasets as the respective outgroup. Black dots indicate strongly supported branches (ultrafast bootstrap ≥ 95 and aLRT SH-like support ≥ 80). Red dots indicate the position of the minimal ancestor deviation (MAD) outgroup-free root in phylogenies without the MurG outgroup (doi.org/10.7910/DVN/9KSWQR). Clades of interest (archaeal glycosylation enzyme 24 [Agl24]-like containing Saci1262, Asgard lineages, and Eukaryotes) are colored. Taxa names are given according to common usage, with a few exceptions where the Genome Taxonomy Database (GTDB) classification was necessary for accuracy. The uncollapsed trees with numerical branch supports are given in **Figure 7—figure supplement 1** and **Figure 7—figure supplement 2**, respectively. The concatenated dataset tree and its uncollapsed version are given in **Figure 7—figure supplement 3** and **Figure 7—figure supplement 4**, respectively.

The online version of this article includes the following figure supplement(s) for figure 7:

**Figure supplement 1.** Uncollapsed phylogeny of Alg13/EpsF.

**Figure supplement 2.** Uncollapsed phylogeny of Alg14/EpsE.

**Figure supplement 3.** Phylogeny of concatenated Alg14-13/EpsEF.

**Figure supplement 4.** Uncollapsed phylogeny of concatenated Alg14-13/EpsEF.

**Figure supplement 5.** Universal distribution of archaeal glycosylation enzyme 24 (Agl24) homologs in Crenarchaeota, except for the Order Thermoproteaces.

The sister clade of Agl24 consists of the eukaryotic Alg13 and Alg14, as well as EpsF-like and EpsE-like sequences from Archaea and Bacteria. With the exception of *Perkinsela* sp. in Alg13/EpsF, Eukaryotes are monophyletic and form a strongly supported monophyletic branch with an Asgard lineage, Thorarchaeota for Alg13 and Odinarchaeota for Alg14. Other closely related archaeal sequences are mainly from Verstraetearchaeota. These, along with the EpsE/EpsF archaeal sequences (mainly from Euryarchaeota and DPANN) are interspersed with bacterial homologs indicating multiple interdomain transfers. The function of EpsE and EpsF has been studied in Lactobacillales, where these enzymes are involved in exopolysaccharide production (**Kolkman et al., 1997**; **De Vuyst et al., 2001**). Similar to Alg14-13, the combined activity of EpsE and EpsF links either a glucose (Glc) from UDP-Glc (**Kleerebezem et al., 1999**, **van Kranenburg et al., 1999**), or a galactose (Gal) from UDP-Gal via a β-1,4 linkage to lipid-linked glucose acceptors (**van Kranenburg et al., 1999**), creating either a lipid-linked cellobiose or a lactose, respectively. As opposed to Agl24 and the other exclusively archaeal clades in our phylogeny, most EpsE/EpsF and Alg14/Alg13-related genes are split. This indicates a single gene split event at the base of that clade, followed by multiple independent re-fusion events across

Bacteria, Archaea (e.g., some Thorarchaeota), and Eukaryotes (Kinetoplastids, Amoebozoa; in the latter with gene order inversion).

All other sequences are exclusively archaeal, fused, and form primarily two clades. One clade consists mainly of euryarchaeal methanogens and a couple of DPANN. The distribution in certain lineages (Methanococcales, Methanosarcinales) is wide enough to suggest that their respective ancestors possessed an Alg14-13 homolog. The second clade corresponds to the TACK superphylum (Thaum-, Bathy-, Aig-, Mars-, Geotherm-, Nezha-, Verstraete-, and Korarchaeota). Even though the relationships among lineages are unresolved, an Alg14-13 homolog was most probably present at the ancestor of the TACK supergroup. Among the TACK, there exist sequences from various Asgard lineages that have been acquired from ancient lateral transfers.

A third clade consisting of divergent Thaumarchaeota sequences is either sister to both TACK and Euryarchaeota-DPANN (Alg13/EpsF, *Figure 7A*) or only the TACK. Although we used only well-established Alg14-13 homologs from *Lombard, 2016*, in our phylogenies, such divergent homologs (often forming distinct clades) exist across multiple archaeal lineages, including one in Bathyarchaeota (mainly in GTDB genus g__PALSA-986) and two in Methanobacteriales (doi.org/10.7910/DVN/9KSWQR). Some of the Bathyarchaeota and the GTDB o__Thermofilales (NCBI: unclassified Thermoprotei) member B67-G16 in the TACK clade above additionally possess an Agl24. Some Bathyarchaeota MAGs also contain additional sequences from the unused divergent clades (*Figure 7—figure supplement 1*, *Figure 7—figure supplement 2*, and doi.org/10.7910/DVN/9KSWQR). Since their N-glycosylation processes are unknown, it is impossible to predict the function and overlap thereof of these homologs.

Outgroup-free rooting with the minimal ancestor deviation (MAD) method (*Tria et al., 2017*) corroborates the outgroup rooting of the phylogenies with MurG (*Figure 7*). The bacterial MurG and various (primarily) archaeal Agl24/Alg14-13/EpsEF homologs seem to have originated from a split at the Last Universal Common Ancestor, as previously proposed (*Lombard, 2016*). Although EpsE/EpsF and Alg14/Alg13-like sequences are found in Bacteria which are ancient in some lineages (EpsE/EpsF in Actinobacteria and Cyanobacteria, see the preliminary phylogenies in doi.org/10.7910/DVN/9KSWQR), these homologs are of archaeal origin and were acquired through ancient interdomain transfers.

Similarly, the respective eukaryotic branches are internally poorly resolved and do not even recover the major clades of that domain, such as Opisthokonts. We cannot reveal if this is purely due to weak phylogenetic signal or horizontal gene transfer events, however, the eukaryotic Alg14-13 are monophyletic and archaeal in origin. One possibility is that they were present in the last eukaryotic common ancestor (LECA), as proposed in *Lombard, 2016*. The proximity of Asgards to Eykaryotes would suggest that these genes were inherited vertically during eukaryogenesis in a two-domain tree of life (*Eme et al., 2017*; *Williams et al., 2020*). The counterargument is that only two Asgard phyla are in the vicinity of Eukaryotes, and neither is Heimdallarchaeota, as would be expected from the literature. Despite our careful analysis, we did not find Alg13- and Alg14-like sequences in most Asgard species, suggesting that if they produce the same type of glycosylation, they could be using a different enzyme for this step of N-glycosylation. The homologs in Wukong-, Baldr-, Borrarchaeota, DTBI01, and a second homolog in Odinarchaeota have been laterally acquired from within the TACK. It is unknown whether the multiple Odinarchaeota genes have overlapping functions. The alternative scenario to eukaryogenesis involves a very ancient interdomain transfer from Archaea to either proto-Eukaryotes or shortly after the LECA. Even if such a history is compatible with both a two- and three-domain tree of life, our phylogenies do not directly support a three-domain tree. Whether with MurG outgroup rooting or MAD outgroup-free rooting, the Eukaryotes emerge from and are not sister to all Archaea.

## Discussion

In this study we identified the first archaeal and thermostable β-1,4-*N*-acetylglucosaminyltransferase, named Agl24, responsible for the second synthesis step of the lipid-linked chitobiose on which the N-glycan of *S. acidocaldarius* is assembled. The reported essentiality of Agl24 is in agreement with the indispensable properties of the OST AglB and AglH, catalyzing the last and first step of the N-glycosylation process in *S. acidocaldarius*, respectively (*Meyer and Albers, 2014*; *Meyer et al., 2017*). The essentiality of the N-glycosylation has also been confirmed by a transposon study in *Sulfolobus islandicus*, a close relative of *S. acidocaldarius*, in which all homologous genes, including *agl24*, lack

any transposon insertions (*Zhang et al., 2018*). However, the fundamental nature of archaeal N-glycosylation cannot be generalized since deletion mutants of the OST AglB can be obtained in Euryarchaeota, for example, in *Methanococcus maripaludis* and *Haloferax volcanii* (*Chaban et al., 2006*; *Abu-Qarn et al., 2007*). The requirement for N-glycosylation might rely on many factors including the N-glycan composition and structure, as well as the modification frequency on proteins; this frequency is reported to be higher in thermophilic, compared to mesophilic, Archaea (*Meyer and Albers, 2013*; *Jarrell et al., 2014*).

Comparison of Agl24 with eukaryotic Alg14-13 and the bacterial MurG enzymes revealed structural and amino acid similarities. Although the sequence identity of Agl24 with the eukaryotic and bacterial enzymes is extremely low (~16%), the structural prediction of Agl24 indicated a MurG-like GT-B fold. Interestingly, the amino acids of the nucleotide sugar donor-binding site of MurG (*Hu et al., 2003*) are not conserved in Agl24 and Alg14, although all enzymes use UDP-GlcNAc as the sugar donor (*Figure 1—figure supplement 2*, boxed). Strikingly, a conserved $GGH_{14}$ motif within an N-terminal loop of MurG, Alg14-13, and Agl24 (*Figure 1* and *Figure 1—figure supplement 2*), which protrudes into the cavity formed by the two structural folds, could be identified in Agl24. In Bacteria, this $G_{13}GTGGH_{18}$ loop is extended by a second $G_{103}GY_{105}$ loop. As both loops are suggested to be reminiscent of the phosphate-binding loops of nucleotide-binding proteins, their involvement in the binding of the diphosphate group of the acceptor lipid has been proposed (*Crouvoisier et al., 2007*). In agreement with our results, the alanine substitutions of the invariant bacterial $H_{14}$ led to an extremely low or undetectable activity of MurG as well as to the loss of the ability to complement a temperature-sensitive MurG mutant (*Hu et al., 2003*; *Crouvoisier et al., 2007*). However, up to now, no structure of MurG or Alg14-13 with the bound lipid acceptor molecule has been published and the interaction of the imidazole ring of the histidine residue with the acceptor molecule remains to be shown. Our alignment also reveals a conserved glutamic acid in the bacterial, eukaryotic, and archaeal sequences. In Bacteria, this glutamic acid is found in a conserved $H_{124}EQN_{127}$ loop (*E. coli* MurG), which has been proposed to form a key part of the acceptor-binding site (*Hu et al., 2003*). In agreement with our study, an $E_{125}A$ substitution significantly affected the function of MurG, although a marked decrease in lipid acceptor binding (1760-fold) and nucleotide substrate binding (210-fold) was observed (*Hu et al., 2003*; *Crouvoisier et al., 2007*).

The euryarchaeal N-glycosylation processes have been well characterized for different species (reviewed in *Jarrell et al., 2014*; *Albers et al., 2017*). The enzymes catalyzing the first and second step, for example, in *H. volcanii* AglJ and AglG (*Yurist-Doutsch et al., 2008*; *Kaminski et al., 2010*) and in *M. voltae* AglK and AglC (*Chaban et al., 2009*; *Larkin et al., 2013*), belong to the GT2 family and possess a GT-A fold. *S. acidocaldarius* is the only crenarchaeal species for which the N-glycosylation process has been defined, and we revealed that Agl24 is the first archaeal glycosylation enzyme with a predicted GT-B structure. Based on these Agl24 differences to the GT2 family and MurG (GT28, GT-B fold), Agl24 is assigned to the new GT-B fold GT-family 115.

Although lacking a TMD, Agl24 associates with the membrane, indicating an interaction with membrane enzymes, similar to the interaction of MurG or Alg14-13 with their membrane interaction partners MraY and Alg7, respectively. Based on the structural similarities of these complexes, a shared common evolutionary origin of the first enzymes of eukaryotic N-glycosylation and the bacterial peptidoglycan biosynthesis has been hypothesized (*Bugg and Brandish, 1994*; *Burda and Aebi, 1999*; *Bouhss et al., 2008*). As *S. acidocaldarius* possesses a homolog of Alg7, termed AlgH (*Meyer et al., 2017*), and a bioinformatics study has indicated that Alg7 might have emerged out of the Crenarchaeota (*Lombard, 2016*), AglH presents the most likely candidate for being the interaction partner of Agl24.

Our phylogenetic analysis shed light on the distribution of the identified Agl24 in Archaea and determined its relationship with eukaryotic and bacterial homologs suggesting a very ancient and broad distribution of Agl24/Alg14-13/EpsEF-like homologs in Archaea. Importantly, close homologs of Agl24 are only found within the Crenarchaeota, Bathyarchaeota, and one Asgard genome (Baldrarchaeota). Indeed, all Crenarchaeota, except for members of the order Thermoproteales, possess an Agl24 homolog (*Figure 7—figure supplement 5*), strongly suggesting that they contain a chitobiose N-glycan core structure. By contrast, the N-glycans of members of the Thermoproteales could be synthesized using a different enzyme. The N-glycan of *Pyrobaculum caldifontis*, belonging to the order Thermoproteales, differs from the N-glycan in Sulfolobales (*Fujinami et al., 2017*). Its

N-glycan core is composed of two di-N-acetylated β-1,4-linked GlcNAc residues with the second sugar being carboxylated at C6 (GlcA(NAc)$_2$-β-1,4-Glc(NAc)$_2$). A di-N-acetylated glucuronic acid sugar is also found in the N-glycan from *M. voltae,* which is transferred by the GT AglC onto the Dol-P-linked GlcNAc (*Chaban et al., 2009*). Thus, a homolog of AglC is likely to be involved in the N-glycan biosynthesis in *P. calidifontis*. The best candidate for an AglC homolog in *P. calidifontis* is Pcal_0481 showing 29.91% sequence identity. Interestingly, the GT-A fold structure has recently been solved, while the predicted function as a mannosyltransferase could not be experimentally confirmed (*Gandini et al., 2020*).

Our phylogenetic analysis revealed that the most closely related enzymes to the eukaryotic Alg14 and Alg13 are found within the Asgard superphylum. Since the discovery of the Archaea as the third domain of life along with Bacteria and Eukaryotes (*Woese and Fox, 1977*), the comparison of archaeal molecular processes, especially those found in Crenarchaeota, has gradually revealed a strong resemblance of the ones found in Eukaryotes (*Huet et al., 1983*; *Zillig et al., 1989*; *Akıl and Robinson, 2018*, *Akıl et al., 2020*). Furthermore, some recent phylogenomic analyses indicated that Eukaryotes originated close to the TACK superphylum (Thaumarchaeota, Aigarchaeota, Crenarchaeota, and Korarchaeota) within the Asgard superphylum (*Petitjean et al., 2014*, *Raymann et al., 2015*; *Zaremba-Niedzwiedzka et al., 2017*; *Williams et al., 2020*; *Liu et al., 2021*) although phylogenies supporting a three-domain tree of life are reported occasionally (*Da Cunha et al., 2017*). A current hypothesis for the formation of the first eukaryotic cell proposes a membrane protrusion of an ancestral archaeon that engulfed a bacterium to increase the interspecies interaction surface (*Baum and Baum, 2014*; *Imachi et al., 2020*). The eukaryotic N-glycosylation would therefore have been inherited from this ancient archaeal ancestor, which is in agreement with the high conservation of the initial N-glycosylation biosynthesis steps in all eukaryotes (*Helenius and Aebi, 2004*; *Samuelson et al., 2005*; *Schwarz and Aebi, 2011*). In addition, AglB in members of the Asgard and TACK superphyla shows considerable similarity to its eukaryotic homologue Stt3, and is clearly distinct from the AglB of the DPANN superphylum and the Euryarchaeota (*Nikolayev et al., 2020*). Most eukaryotic Stt3 contain a double sequon motif for N-glycosylation near the substrate-binding site, which is hypothesized to contribute to the LLO (*Wild et al., 2018*; *Shrimal and Gilmore, 2019*). Interestingly, this double sequon is also present in the AglB of the members of the Asgard and TACK superphyla but absent in the AglB of Euryarchaeota (*Shrimal and Gilmore, 2019*). Moreover, homologs of non-catalytic subunits of the eukaryotic Stt3 complex, for example, ribophorin, Ost3/6, Ost5, and Wbp1, have been detected in some Asgard (*Zaremba-Niedzwiedzka et al., 2017*). However, their specific function in the N-glycosylation has to be shown, especially as ribophorin is only found in mammals (*Wilson and High, 2007*) and is therefore unlikely to have existed in early Eukaryotes. Nevertheless, the assembly of the N-glycan linkage in the TACK superphylum is conducted via a pyrophosphate linkage to the Dol (*Taguchi et al., 2016*), identical to that in Eukaryotes. This contrasts with Euryarchaeota, which use a monophosphate linkage. A summary of the similarities and differences of the archaeal N-glycosylation to Eukaryotes and Bacteria is given in *Supplementary file 4*. All these observations strengthen the hypothesis that the eukaryotic N-glycosylation has emerged from an ancient archaeon. However, an origin of the eukaryotic Alg14-13 through eukaryogenesis cannot be unambiguously demonstrated. Due to the sparse distribution of Alg14-13 homologs among Asgards, incongruences in the phylogenies of the two proteins (*Figure 7*), and poor resolution within Eukaryotes (*Figure 7—figure supplement 1* and *Figure 7—figure supplement 2*), it is possible that these components of eukaryotic N-glycosylation originated from an ancient lateral transfer. It is also noteworthy that there exists a multitude of distant Alg14-13 homologs throughout Archaea. Even though in our analysis we relied on well-established clades, primarily due to their evolutionary proximity to Agl24 and Eukaryotes, there are additional clades other than the Asgards worth exploring in the future, such as the ones in Bathyarchaeota and Methanobacteriales (doi.org/10.7910/DVN/9KSWQR). Many Bathyarchaeota have multiple Alg14-13 homologs, which suggest some functional redundancy. Methanobacteriales use pseudomurein, which raises the question why do they contain these homologs.

The homologs of eukaryotic GTs in members of the TACK and Asgard superphyla are very useful tools to study the structure and the functional relationship of the elusive eukaryotic enzymes. In particular, *S. acidocaldarius*, as one of the few genetically treatable archaea, produces thermostable GTs, that allow detailed enzymatic and structural characterization.

## Materials and methods

### Strains and growth conditions

The strain *S. acidocaldarius* MW001 (Δ*pyrE*) (*Wagner et al., 2012*) and all derived modified strains (*Supplementary file 3*) were grown in Brock medium at 75°C, pH 3, adjusted using sulfuric acid. The medium was supplemented with 0.1% w/v NZ-amine and 0.1% w/v dextrin as carbon and energy source (*Brock et al., 1972*). Selection gelrite (0.6%) plates were supplemented with the same nutrients, with the addition of 10 mM $MgCl_2$ and 3 mM $CaCl_2$. For the second selection plates, 10 mg $mL^{-1}$ uracil and 100 mg $mL^{-1}$ 5-fluoroorotic acid were added. For the growth of the uracil auxotrophic mutants, 10 mg $mL^{-1}$ uracil was added to the medium. Cell growth was monitored by measuring the optical density at 600 nm. Protein expression in *S. acidocaldarius* was conducted in medium supplemented with 0.1% w/v NZ-amine and 0.1% w/v L-arabinose as carbon and energy source. All *E. coli* strains, DH5α, BL21, or ER1821, were grown in LB media at 37°C in a shaking incubator at 200 rpm. According to the antibiotic resistance in the transformed vector(s), media were supplemented with the antibiotics carbenicillin (amp) at 100 µg $mL^{-1}$ and/or kanamycin (kan) at 50 µg $mL^{-1}$.

### Construction of deletion plasmids

The predicted function of *agl24* was verified via the generation of the lipid-linked chitobiose core of the N-glycan. Therefore, a marker-less deletion mutant of this gene was constructed in *S. acidocaldarius* MW001, as previously described (*Wagner et al., 2012*). Used primers and generated plasmid are given in *Supplementary file 3*, the strain MW001, auxotrophic for uracil biosynthesis, was transformed with the plasmid pSVA1312. Therefore, two ~1000 bp DNA fragments, one from the upstream and one from the downstream regions of *agl24* (*saci1262*) gene, were PCR amplified. Restriction sites *Apa*I and *Bam*HI were introduced at the 5′ ends of the upstream forward primer (4168) and of the downstream reverse primer (4165), respectively. The upstream reverse primer (4163) and the downstream forward primer (4164) were each designed to incorporate 15 bp of the reverse complement strand of the other primer, resulting in a 30 bp overlap stretch. The overlapping PCR fragments were purified, digested with *Apa*I and *Bam*HI, and ligated in the digested plasmid pSVA407, containing *pyrEF* (*Wagner et al., 2012*).

### Generation of a linear *Agl24*<sub>up</sub>-*pyrEF*-*Agl24*<sub>down</sub> fragment for the direct *agl24::pyrEF* replacement

To further underline the essential role of *agl24* in *S. acidocaldarius*, a disruption of the *agl24* gene by direct homologous integration of the *pyrEF* cassette was performed. For this approach, 387 bp of the *agl24* upstream region, the full 1525 bp of the *pyrEF* cassette, and 1011 bp of the *agl24* downstream region were PCR amplified using the primers: 4168 + 6336, 4115 + 4116, and 6337 + 6338, respectively. At the 5′ ends of the upstream forward primer (4168) and of the downstream reverse primer (6338) restriction sites *Apa*I and *Bam*HI were introduced, respectively. The *pyrEF* forward primer (4115) was designed to incorporate 40 bp of the upstream reverse primer (6336) resulting in a 40 bp overlapping stretch. The *pyrEF* reverse primer (4116) and the downstream forward primer (6337) were designed to incorporate the reverse complement strand of the other primer, resulting in a 46 bp overlapping stretch. The upstream, *pyrEF*, and downstream fragments were fused by an overlapping PCR, using the 3′ ends of each fragment as primers. The 2904 bp overlap PCR fragment gained, was amplified using the outer primers (4168 and 6338), digested with *Apa*I and BamHI, and ligated into the p407 vector predigested with the same restriction enzymes. The resulting plasmid, pSVA3338 (*agl24*<sub>up</sub>-*pyrEF*-*agl24*<sub>down</sub>), was used to transform *E. coli* DH5α and selected on LB-plates containing 50 mg $mL^{-1}$ ampicillin. The accuracy of the plasmid was verified by sequencing. Before transformation in *S. acidocaldarius*, the plasmid was digested with *Apa*I and *Bam*HI to create the linear *agl24*<sub>up</sub>-*pyrEF*-*agl24*<sub>down</sub> fragment.

### Generation of *Agl24* expression plasmids for the production of Agl24 in *S. acidocaldarius* and *E. coli*

Cellular localization studies were carried out using N- and C-terminal fusion proteins of Agl24. The *agl24* gene was amplified from genomic DNA of *S. acidocaldarius* introducing *Nco*I and *Pst*I restriction sites at the 5′ ends of the primers 4176 and 4177, respectively. The PCR fragment was cloned

into pSVA1481, yielding vector pSVA1336 containing an inducible arabinose promoter and *agl24-strep-his*$_{10}$. The plasmid was digested with *Nco*I and *Eag*I and the insert *agl24-strep-his*$_{10}$ was ligated into pSVA1481 containing an inducible maltose promoter, yielding pSVA1337. The N-terminal *strep-his*$_{10}$-*tev-agl24* was cloned using the primer pair 4180 and 4181, which incorporated the restriction sites *Nco*I and *Not*I, respectively. The insert was ligated into pSVA2301 containing an inducible maltose promoter yielding pSVA1339.

For heterologous expression of *agl24* in *E. coli,* the plasmid pHD0499 was generated. The GFP-His$_8$-tagged *agl24* gene was constructed by amplifying the full-length *agl24* gene from *S. acidocaldarius* genomic DNA with the primer pair A596 and A597. The PCR fragment was cloned in-frame into the vector pWaldo (*Waldo et al., 1999*) using *Xho*I and *Kpn*I restriction sites, generating a C-terminal fusion with a TEV cleavage site, GFP, and His$_8$ tag. For the generation of Agl24 mutants of the conserved mutants H$_{15}$ and E$_{114}$, a quick-change mutagenesis PCR was applied using the primers: A597 and A684 or A685 and A686, respectively.

## Transformation and selection of the deletion mutant in *S. acidocaldarius*

Generation of competent cells was performed based on the protocol of *Kurosawa and Grogan, 2005*. Briefly, *S. acidocaldarius* strain MW001 was grown to an OD$_{600}$ between 0.1 and 0.3 in Brock medium supplemented with 0.1% w/v NZ-amine and 0.1% dextrin. Cooled cells were harvested by centrifugation (2000× *g* at 4°C for 20 min). The cell pellet was washed three times successively in 50, 10, and 1 mL of ice-cold 20 mM sucrose (dissolved in demineralized water) after mild centrifugation steps (2000× *g* at 4°C for 20 min). The final cell pellet was resuspended in 20 mM sucrose at an OD$_{600}$ of 10.0 and stored in 50 µL aliquots at –80°C; 400–600 ng of methylated pSVA1312 or the linearized *Agl24*$_{up}$-*pryEF-Agl24*$_{down}$ fragment were added to a 50 µL aliquot of competent MW001 cells and incubated for 5 min on ice before transformation in a 1 mm gap electroporation cuvette at 1250 V, 1000 Ω, 25 mF using a Bio-Rad gene pulser II (Bio-Rad, Plano, TX). Directly after transformation, 50 µL of a 2× concentrated recovery solution (1% sucrose, 20 mM β-alanine, 20 mM malate buffer pH 4.5, 10 mM MgSO$_4$) were added to the sample and incubated at 75°C for 30 min under mild shaking conditions (150 rpm). Before plating, the sample was mixed with 100 µL of heated 2× concentrated recovery solution and twice 100 µL were spread onto gelrite plates containing Brock medium supplemented with 0.1% NZ-amine and 0.1% dextrin. After incubation for 5–7 days at 75°C, large brownish colonies were used to inoculate 50 mL of Brock medium containing 0.1% NZ-amine and 0.1% dextrin, which were incubated for 3 days at 78°C. Cultures confirmed by PCR to contain the genomically integrated plasmid were grown in Brock medium supplemented with 0.1% NZ-amine and 0.1% dextrin to an OD of 0.4. Aliquots of 40 µL were spread on second selection plates, supplemented with 0.1% NZ-amine and 0.1% dextrin and 10 mg mL$^{-1}$ uracil, were incubated for 5–7 days at 78°C. Newly formed colonies were streaked on fresh second selection plates to ensure single colony selection before each colony was screened by PCR for the genomic absence, presence, or modification of the *agl24* gene.

## Expression of Agl24 in *S. acidocaldarius*

Plasmid transformation of *S. acidocaldarius* cells was performed as described above for the deletion mutants. Cells were spread onto gelrite plates containing Brock medium supplemented with 0.1% NZ-amine and 0.1% dextrin. After incubation for 5–7 days at 75°C, large brownish colonies were used to inoculate 50 mL of Brock medium containing 0.1% NZ-amine and 0.1% dextrin and incubated for 3 days at 78°C. Presence of the expression plasmid was confirmed by PCR. For induction of expression the strains were grown either in Brock medium supplemented with either 0.1% NZ-amine and 0.1% dextrin (MAL$_{promotor}$) or with 0.1% NZ-amine and 0.1% L-arabinose (ARA$_{promotor}$) to an OD$_{600}$ of 1.0.

## Expression and purification of recombinant Agl24 protein from *E. coli*

For heterologous Agl24 protein expression, 6 × 1 L LB medium was inoculated with 10 mL from an overnight BL21 culture previously transformed with pHD0499. Cells were grown at 16°C overnight in auto-induction medium (*Studier, 2014*) containing 30 µg mL$^{-1}$ kanamycin. The cells were harvested by centrifugation at 4200× *g* for 25 min and used to isolate the membrane fractions. All subsequent purification steps were carried out at 4°C. Cells were fractionated by passing four times through an Avestin C3 High Pressure Homogeniser (Biopharma, UK), followed by a 20 min low speed spin at 4000× *g*. The resulting supernatant was centrifuged at 200,000× *g* for 2 hr to obtain the membrane fraction

and 2–3 g of membranes were routinely used for isolation of Agl24-GFP-His$_8$ proteins. Samples were solubilized in 18 mL Buffer 1 (500 mM NaCl, 10 mM Na$_2$HPO$_4$, 1.8 mM KH$_2$PO$_4$ 2.7 mM KCl, pH 7.4, 20 mM imidazole, 4 mM TCEP) with the addition of 1% *n*-dodecyl-β-maltoside (DDM) for 2 hr at 4°C. The sample was twofold diluted with Buffer 1 and centrifuged at 200,000× *g* for 2 hr. The supernatant was loaded onto a Ni-Sepharose 6 Fast Flow (GE Healthcare) column with 1 mL of prewashed Ni-NTA-beads. The column was washed with 20 mL of wash buffer (500 mM NaCl, 10 mM Na$_2$HPO$_4$, 1.8 mM KH$_2$PO$_4$ 2.7 mM KCl, pH 7.4, 20 mM imidazole, 0.4 mM TCEP, 0.03% DDM) and eluted with 5 × 1 mL elution buffer (500 mM NaCl, 10 mM Na$_2$HPO$_4$, 1.8 mM KH$_2$PO4 2.7 mM KCl, pH 7.4, 250 mM imidazole, 0.4 mM TCEP, 0.03% DDM). Elution fractions were combined, and imidazole was removed using a HiPrep 26/10 desalting column (GE Healthcare) equilibrated with the following Buffer (1× PBS, 0.03% DDM, 0.4 mM TCEP). Protein concentration was determined using a Bradford reaction (Bio-Rad) and purity was confirmed by SDS-PAGE analysis. The concentrated fractions were separated by SDS-PAGE and stained with Coomassie blue. Protein identity was confirmed by tryptic peptide mass fingerprinting, and the level of purity and molecular weight of the recombinant protein was determined by matrix-assisted laser desorption/ionization time-of-flight mass spectrometry (MALDI-TOF). The analysis was provided by the University of Dundee 'Fingerprints' Proteomics Facility.

## Agl24 activity

Activity was measured in 100 µL reaction volume containing 1 mM UDP-D-GlcNAc, 1 mM acceptor molecule, 5 mM MgCl$_2$, and 5 µg of purified Agl24 in TBS Buffer (150 mM NaCl, 50 mM Tris-HCl, pH 7.6). The reaction was performed in a PCR cycler at 60 °C for 12 hr.

## MS analysis

MALDI-TOF was used to analyze the acceptors and products of the Agl24 in vitro assay. At least three repeats were performed, where 100 µL reaction samples were purified over 100 mg Sep-Pak C18 cartridges (Waters) pre-equilibrated with 5% EtOH. The bound samples were washed with 800 µL of H$_2$O and 800 µL of 15% EtOH, eluted in two fractions with (1) 800 µL of 30% EtOH and (2) 800 µL of 60% EtOH. The two elution fractions were dried in a SpeedVac vacuum concentrator and resuspended in 20 µL of 50% MeOH. A 1 µL sample was mixed with 1 µL of 2,5-dihydroxybenzoic acid matrix (15 mg mL$^{-1}$ in 30:70 acetonitrile, 0.1% TFA), and 1 µL was added to the MALDI grid. Samples were analyzed by MALDI in an Autoflex Speed mass spectrometer set up in reflection positive ion mode (Bruker, Germany).

## HPLC analysis

With the exception of kinetics reactions, which are detailed below, Agl24 reactions were analyzed using an HPLC assay with, typically, 50 µL samples containing 2.5 mM UDP-GlcNAc, 1.5 mM lipid acceptor, and 1.8 µg of purified Agl24 in a TBS buffer containing 5 mM MgCl$_2$ (150 mM NaCl, 50 mM Tris-HCl, pH 7.5). Reactions were left at 60°C (or alternative temperatures) for the desired time and quenched with one equivalent of acetonitrile to precipitate Agl24. Following filtration to remove precipitate, reactions were injected onto an XBridge BEH Amide OBD Prep column (130 Å, 5 µM, 10 × 250 mm) at a flow rate of 4 mL min$^{-1}$ using a Dionex UltiMate 3000 system (Thermo Scientific) fitted with a UV detector optimized to detect the *O*-phenyl functional group of the acceptor molecule at 270 nm. Each run was 35 min using running Buffer A (95% acetonitrile, 10 mM ammonium acetate, pH 8) and Buffer B (50% acetonitrile, 10 mM ammonium acetate, pH 8). A linear gradient from 20% to 80% Buffer B was performed over 20 min, followed by an immediate drop back to 20% Buffer B for the remaining 15 min of the run to re-equilibrate the column to starting conditions. The Agl24 reaction substrates and products typically eluted 8–11 min into a run. For kinetic analyses, assays were performed in triplicate (with the exception of 1 mM concentration which was only performed in duplicate) and altered to contain a fixed concentration of lipid acceptor (0.5 mM) while varying the concentration of UDP-GlcNAc (0.5, 0.75, 1, 1.5, 3, 6, 9, 12 mM). Reactions were run for 10 hr before being quenched and, after completion, areas of substrate and product peaks were calculated to determine the reaction conversion. Conversion over 10 hr was converted to pmol per minute, and the resulting data were analyzed using GraphPad Prism v8 to generate a Michaelis–Menten curve and the resulting kinetic data.

## NMR analysis

The HPLC-purified Agl24 products from three to five reactions each (0.5–2 mg) were combined and dried using a Christ RVC 2–25 speed vacuum fitted with a Christ CT 02–50 cold-trap to remove excess acetonitrile, then freeze dried (Alpha 1–2 LDplus, Christ) to remove residual water. Products were subsequently dissolved in 600 µL of $D_2O$ and NMR spectra were recorded at 293 K. The spectra were acquired on a Bruker AVANCE III HD 500 MHz NMR Spectrometer equipped with a 5 mm QCPI cryoprobe. For 1D TOCSY experiments, H1′ was irradiated at 5.35 ppm, H2′ was irradiated at 3.90 ppm, and H1″ was irradiated at 4.49 ppm (*Figure 6—figure supplement 1*). A combination of $^1H$ $^1H$ COSY, 1D TOCSY, and $^1H$ $^{13}C$ HSQC experiments were used to fully assign the $^1H$ and $^{13}C$ signals for the Agl24 reaction product. Full $^1H$ and $^{13}C$ chemical shift assignments can be found in *Table 1* and are recorded with respect to the residual HDO signal at 4.7 ppm.

## BLAST searches and homology modeling of Saci1262

To identify homologs of MurG (WP_063074721.1), Alg13 (NP_011468.1), Alg14 (NP_009626.1), and Alg14-13 fusion (NP_009626.1 + NP_011468.1) in *S. acidocaldarius,* a DELTA BLAST (https://blast.ncbi.nlm.nih.gov/Blast.cgi) was performed with the standard parameters (*Boratyn et al., 2012*). Homology modeling of Saci1262 (Agl24) was performed on the SWISS-MODEL server (*Waterhouse et al., 2018*) with default parameters.

The obtained sequences of the MurG, Alg14-14, and Saci1262 homologs were alignment with Clustal Omega (*Sievers and Higgins, 2018*) with standard parameter and visualized with Geneious Prime 2020 (Biomatters, Ltd. New Zealand) with an amino acids conservation threshold of 65%.

## Phylogenetic analyses

To search for homologs of Alg14-13 in Archaea, Bacteria, and Eukaryotes, we constructed a local database for each domain. The Archaea database (2424 genomes) contained all representative genomes from GTDB r202 (*Parks et al., 2022*), plus the Asgard genomes from *Liu et al., 2021* (NCBI BioProject identifier PRJNA680430). We downloaded them as nucleotide contigs with ncbi-genome-download (https://github.com/kblin/ncbi-genome-download; (*Meyer, 2021* copy archived at swh:1:rev:79b71d322f7194ccc44aa3b543a05c38ad271cfb)) and predicted protein sequences with Prokka 1.14.0 (*Seemann, 2014*). For Bacteria (25,118 genomes), we used all entries on NCBI Genome as of 2019/06/01, dereplicated at species level through a combination of rpS3 clustering and dRep (*Olm et al., 2017*). We then downloaded the genomes from NCBI as protein fasta trying first GenBank, then RefSeq, and finally GenBank nucleotide contigs followed by predicting protein sequences with Prodigal (*Hyatt et al., 2010*). For Eukaryotes (1611 genomes), we used all entries on NCBI Genome as of 2019/06/01, taking one genome per species based on the txids using the completion status to pick said genome (in order of preference: Reference, Representative, Complete, Chromosome, Scaffold, and finally Contig state).

For all homology searches we used DIAMOND v2.0.8.146 (*Buchfink et al., 2021*) with the default e-value cutoff (--outfmt 6 –ultra-sensitive –max-target-seqs 0). For Archaea we used one query sequence for each of the main clades from *Lombard, 2016*: (1) the *S. acidocaldarius* Agl24 from this study; (2) the Alg13/EpsF-like and Alg14/EpsE-like homologs from *Methanothrix soehngenii* (NCBI: AEB67939.1 and AEB67938.1, respectively); (3) AIC14646.1 from *Nitrososphaera viennensis* (TACK clade); (4) ABX11947.1 from *Nitrosopumilus maritimus* (additional Alg13/EpsF-like Thaumarchaeota clade); (5) AAB99267.1 from *Methanocaldococcus jannaschii* (methanogen clade). For Bacteria and Eukaryotes, we used all non-MurG sequences from the corresponding Alg13/EpsF and Alg14/EpsE datasets from *Lombard, 2016*. To isolate the homologs of interest, we aligned these datasets with MAFFT FFT-NS-2 v7.453 (*Katoh and Standley, 2013*) (--reorder), trimmed the alignments with ClipKIT (*Steenwyk et al., 2020*) (-m kpi-gappy -g 0.5) and constructed phylogenies with IQ-TREE 2.0.5 (*Minh et al., 2020*) (-m TEST -mset JTT,WAG,LG -bb 1000 -alrt 1000) with 1000 each ultrafast bootstrap (*Hoang et al., 2018*) and aLRT SH-like (*Guindon et al., 2010*) replicates.

The homologs of interest were then isolated by manually inspecting the phylogenies and alignments. For Archaea we took all sequences associated with the clades from *Lombard, 2016*, using the same fused (non-EpsE/EpsF) homologs in both the Alg13/EpsF and Alg14/EpsE datasets (e.g., this resulted in adding the additional Thaumarchaeota sequences to the Alg14/EpsE dataset, whereas in *Lombard, 2016*, they were only found in the Alg13/EpsF dataset). The *M. kandleri* sequence that

formed a divergent clade in the (*Lombard, 2016*) Alg14/EpsE trees was omitted altogether. For Eukaryotes we picked representative sequences trying to cover their known taxonomic range from the literature. For Bacteria we tried to gather representatives from all non-MurG-like clades of the preliminary phylogenies (see doi.org/10.7910/DVN/9KSWQR). For both Bacteria and Eukaryotes we tried to match the taxa contained in the Alg13/EpsF and Alg14/EpsE datasets based on the preliminary phylogenies and synteny (doi.org/10.7910/DVN/9KSWQR), in order to be able to concatenate sequences from split genes (see below). As outgroups, we used the bacterial MurG sequences from the respective (*Lombard, 2016*) datasets. The final datasets were realigned with MAFFT E-INS-i and trimmed with ClipKIT as above. The final phylogenies were inferred with IQ-TREE (-m MFP -mset JTT,WAG,LG -bb 1000 -alrt 1000) under the model determined by ModelFinder (*Kalyaanamoorthy et al., 2017*) and branch support tests as above. In spite of incongruencies between the Alg13/EpsF and Alg14/EpsE phylogenies, we also created pseudo-concatenated datasets by fusing all split sequences with a custom Python script (https://github.com/ProbstLab/Adam_Kolyfetis_2021_methanogenesis; *Adam et al., 2021*) in the order Alg14-13/EpsEF, as commonly found is archaeal Agl24-like and bacterial MurG homologs. For eukaryotic sequences that have resulted from gene fusion events (see Results), we checked the gene order through each sequence's Uniprot cross-reference to Pfam. Whenever the genes were fused inversely (i.e., Alg13-14), we reversed them for the concatenation by manually taking the part of the sequence from the N-terminal to the end of Alg13 (Glyco_tran_28_C) as per the cross-reference and moving it to the end of the sequence (the C-terminal of Alg14). The concatenated dataset was aligned, trimmed, and a phylogeny was run as above. All trees were visualized in iTol (*Letunic and Bork, 2019*). Outgroup-free rooting was performed with the MAD method (*Tria et al., 2017*).

## Acknowledgements

We thank Vladimir I Torgov, Vladimir V Veselovsky, and Leonid L Danilov from the ND Zelinsky Institute of Organic Chemistry, Russian Academy of Sciences, Moscow, Russia, for sharing the synthetic acceptor molecules. We thank Mr Tom Snelling for assay assistance. Funding: The HCD laboratory (BHM, BAW, and HCD) is supported by The Wellcome Trust and Royal Society Grant 109357/Z/15/Z and the University of Dundee Wellcome Trust Fund 105606/Z/14/Z. SVA and BHM have been supported by the SFG grant SFB 1381 (Deutsche Forschungsgemeinschaft (German Research Foundation) under project no. 403222702-SFB 1381). PSA is supported by a postdoctoral fellowship from the Alexander von Humboldt Foundation. AJP is supported by funding from the Ministerium für Kultur und Wissenschaft des Landes Nordrhein-Westfalen ('Nachwuchsgruppe Dr Alexander Probst'). For the purpose of Open Access, the authors have applied a CC BY public copyright license to any Author Accepted Manuscript version arising from this submission.

## Additional information

### Funding

| Funder | Grant reference number | Author |
|---|---|---|
| The Wellcome Trust and Royal Society Grant | 109357/Z/15/Z | Benjamin Meyer<br>Ben A Wagstaff<br>Helge C Dorfmueller |
| SFG | 403222702-SFB 1381 | Benjamin Meyer<br>Sonja V Albers |
| Ministerium für Kultur und Wissenschaft des Landes Nordrhein-Westfalen | Nachwuchsgruppe Dr. Alexander Probst | Alexander J Probst |
| Alexander von Humboldt Foundation | Postdoctoral fellowship | Panagiotis S Adam |

The funders had no role in study design, data collection and interpretation, or the decision to submit the work for publication.

## Author contributions
Benjamin H Meyer, Conceptualization, Data curation, Formal analysis, Investigation, Methodology, Project administration, Validation, Visualization, Writing – original draft, Writing – review and editing; Panagiotis S Adam, Data curation, Formal analysis, Investigation, Methodology, Supervision, Validation, Visualization, Writing – review and editing; Ben A Wagstaff, Data curation, Formal analysis, Investigation, Methodology, Validation, Visualization, Writing – review and editing; George E Kolyfetis, Data curation, Formal analysis, Investigation, Visualization, Writing – review and editing; Alexander J Probst, Formal analysis, Investigation, Methodology, Supervision, Validation, Visualization, Writing – review and editing; Sonja V Albers, Conceptualization, Data curation, Funding acquisition, Methodology, Project administration, Resources, Supervision, Validation, Writing – review and editing; Helge C Dorfmueller, Conceptualization, Data curation, Formal analysis, Funding acquisition, Methodology, Project administration, Resources, Supervision, Validation, Writing – original draft, Writing – review and editing

## Author ORCIDs
Benjamin H Meyer (iD) http://orcid.org/0000-0002-4986-9511
Panagiotis S Adam (iD) http://orcid.org/0000-0003-4251-7262
Sonja V Albers (iD) http://orcid.org/0000-0003-2459-2226
Helge C Dorfmueller (iD) http://orcid.org/0000-0003-1288-044X

## Decision letter and Author response
Decision letter https://doi.org/10.7554/eLife.67448.sa1
Author response https://doi.org/10.7554/eLife.67448.sa2

---

# Additional files

## Supplementary files
• Transparent reporting form

• Supplementary file 1. Results of the Domain Enhanced Lookup Time Accelerated BLAST (https://blast.ncbi.nlm.nih.gov/Blast.cgi). DELTA BLAST was performed to identify homologs of MurG (WP_063074721.1), Alg13 (NP_011468.1), Alg14(NP_009626.1), and Alg14-13 fusion in Sulfolobus acidocaldarius.

• Supplementary file 2. Structural alignment modeling of Agl24 revealed a conservationof fold to the available structures of MurG and Alg13. Structural modelling was performed by SWISS-MODEL (Waterhouse et al., 2018), using either the full length Agl24 sequence or only the c-terminal Alg13-like part. Identified PDB numbers as well as the SWISS MODLE results, e.g.Global Model Quality Estimation (GMQE), quaternary structure quality estimate (QSQE), and sequence identity are shown.

• Supplementary file 3. Strains, plasmids, and Primers used in this study.

• Supplementary file 4. Overview of the properties of the archaeal, eukaryal andbacterial N-glycosylation. Highlighted in grey background color are the mammalian orthologs of the non-catalytic OST subunits found in *Saccharomyces cerevisiae*. *based on sequence analyses of metagenome-assembled genomes of Asgardarchaeota (Zaremba-Niedzwiedzka et al.,2017), **not present in STT3 proteins from organisms that express single subunit OSTs. ***chitobiose is not present in Thermoproteales, but showing an modified version: $(GlcA(NAc)_2-\beta-1,4-Glc(NAc)_2)$.

## Data availability
Seed sequences, homology searches, preliminary phylogenies, final datasets (individual homologs), final datasets (full and trimmed alignments), and final phylogenies for the phylogenetic analysis are collected in a zip file, which can be assess under the following DOI: https://doi.org/10.7910/DVN/9KSWQR.

The following dataset was generated:

| Author(s) | Year | Dataset title | Dataset URL | Database and Identifier |
|---|---|---|---|---|
| Meyer BH | 2022 | Supplementary_Data-1-Agl24 | https://doi.org/10.7910/DVN/9KSWQR | Harvard Dataverse, 10.7910/DVN/9KSWQR |

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
