## [Editor Report]

In this manuscript, Meyer and co-workers further contribute to our understanding of how proteins are modified with carbohydrates on asparagine residues, N-glycosylation. They analyze this in the thermoacidophilic archaea Sulfolobus acidocaldarius and characterize Agl24 as a key player in the N-glycosylation pathway in this organism. These findings advance the glycobiology field and reveal how distinct mechanisms have arisen across evolution to post-translationally modify protein structures.

---

## [Decision Letter]

**Decision letter after peer review:**

Thank you for submitting your article "N-glycan chitobiose core synthesis by Agl24 strengthens the hypothesis of an archaeal origin of eukaryal N-glycosylation" for consideration by *eLife*. Your article has been reviewed by 3 peer reviewers, and the evaluation has been overseen by a Reviewing Editor and Michael Marletta as the Senior Editor. The following individual involved in review of your submission has agreed to reveal their identity: Patrick Forterre (Reviewer #2).

Essential revisions:

1) Revise text for accuracy and readability as noted by the reviewers. Revise title, summary, discussion, and cover art to align with data provided.

2) Include statements to make the impact of the work on glycobiology more clear for the reader. Strengthen the messaging of the study as it relates to areas of emphasis noted by the reviewers (use of model system, motif, etc).

3) Revise the phylogenetic analysis to strengthen it.

4) Figure S4. Provide a control showing the subcellular fraction of another S. acidocaldarius protein of known distribution, to confirm the membrane association of Agl24. If authors are unable to provide this, then the text should be edited appropriately to make the conclusions tentative pending further confirmation in line with evidence provided.

*Reviewer #1 (Recommendations for the authors):*

General comment – While the actual experiments are solid, the manuscript is very poorly written and even sloppy at places. Too many imprecise statements, errors in figures, etc (see examples, below). The manuscript would also be improved by a quick round of English language editing.

My specific comments are listed as encountered while reading the manuscript:

Introduction

Page 1, line 26 (Cover Art) – in the upper left-hand corner, an outline of something remains. Remove it.

Page 2, line 29 (and Page 2, line 54) – In Bacteria, N-glycosylation is seemingly limited to the δ/epsilon proteobacteria. As such, the statement: "Protein N-glycosylation is the most common posttranslational modifications found in all three domains of life" is inaccurate. Indeed, the authors themselves note this point (page 3, line 66).

Page 2, line 47 – remove comma between "histidine" and "within".

Page 2, line 55 – does *eLife* require references to be listed alphabetically or chronologically? Here, neither rule applies. Add a period at the end of the sentence.

Page 2, line 56 – the text reads: "In Eukarya, N-glycosylation is an essential process", yet on line 61, we are told: "the biological functions of protein glycosylation range from relatively minor to crucial for survival". The two phrases don't match.

Page 3, line 69 – what about the non-lipid-dependent N-glycosylation that occurs in some bacteria (e.g. Haemophilus influenzae HMW1 adhesin or the actions of NGT, the N-glycosylating GTase in Actinobacillus pleuropneumoniae )?

Page 3, line 74 – to the best of my knowledge, the extended bacterial sequon D/E-X1-N-Χ2-S/T (where X is not proline) is recognized in Campylobacter jejuni and Helicobacter pullorum, the OST of other bacteria show a more relaxed specificity (for instance, see Ollis et al., 2015 Sci Reports).

Page 3, line 77 – The statement: "Eukarya require an OST complex of nine subunits" is not accurate. Trypanosomes use a single subunit-containing OST.

Page 3, lines 78-82 – The writing here is very sloppy. In lines 78-79, we are told: "In contrast, Archaea use the shorter version of the recognition sequon (N-X-S/T) as well as dolichol-phosphate (Dol) as lipid", yet lines 81-82 read: "while the Crenarchaeota use Dol pyrophosphate as in eukaryotes".

Page 4, line 103 – "This is a rare example of cross-domain complementation" is inaccurate. There are numerous examples of complementation of a eukaryote mutant by a bacterial gene (for instance, see Kachroo et al., *eLife* 2017 for examples). Moreover, no hypothesis of a common origin for eukaryotic and archaeal N-glycosylation was stated up to this point, so it is not clear how this statement further strengthens such a concept.

Page 4, lines 108-109 – Agl24 would not generate the lipid-linked chitobiose core; rather, it would contribute to the assembly of this moiety.

Page 4, line 110 – On lines 103-104, the reader is told "of a common origin of eukaryotic and archaeal N-glycosylation", yet here it is proposed that "the eukaryotic N-glycosylation pathway originates from an archaeal ancestor". These scenarios are not really the same thing.

Larkin et al., (2013, Nat Chem Biol), where AglK was shown to add GlcNAc to dolichol phosphate in the methanoarchaea Methanococcus voltae should also be addressed in the Introduction, especially given how genetic studies on this organism have assigned this role to AglH (Shams-Eldin et al., 2008, J. Bacteriol.).

Likewise, the N-linked glycan of the crenarchaeote Pyrobaculum calidifontis, which includes a core reminiscent of chitobiose (Fujinami et al., 2017, Glycobiology) should also be noted in the Introduction.

Results

Page 5, line 115 – "this"?

Page 5, lines 123-124 – Not accurate. CAZy lists the halophile Salinadaptatus halalkaliphilus, a member of the family Natrialbaceae, as a GT28 family member (http://www.cazy.org/GT28_archaea.html). Also, Kandler and/or Konig should be cited for their work on pseudomurein.

Page 5, line 142 – The authors should refer to the paper by Cavalier-Smith and Chao (2020, Protoplama) where Saci1262 is described as corresponding to MurG. Indeed, this reference addresses similarities between MurG, Alg13/14 and Saci1262.

Page 6, line 148 – At this stage of the story, there is no justification in assigning Saci1262 to the N-glycosylation pathway, namely, assigning it the Agl tag used for confirmed archaeal N-glycosylation pathway components. Such involvement needs to be first demonstrated – no such demonstration has been presented at this stage of the report.

Page 6, lines 149-153 – Since the reader is not told where other N-glycosylation genes are found relative to Saci1262, this bit of text could confuse those outside the field. Indeed, the authors finish the section by stating how Saci1262 is only eight genes downstream of the OST-encoding aglB gene, leaving the reader unclear as to how N-glycosylation gene clustering is defined.

The "Detailed bioinformatics….." section beginning on line 156 would be easier to read if the authors better separated data based on sequence comparisons from data based on structural comparisons. Moreover, including evolutionary considerations in the middle of the first paragraph of the section (lines 163-168) further adds to mixture of ideas that require better organization.

Page 6, line 170 – "archaeal Agl24 enzymes"? At this point, the reader has only been told of Saci1262. Figure 1B and Figure S2, where archaeal Saci1262 homologues are listed, have not yet been cited in the text. Indeed, without knowing anything about N-glycosylation in the majority of these species, it is premature to assign them the Agl24 title – at this stage they are homologues of Saci1262.

Figure 1A – "Archaea" should not be italicized. Also, why "most Eukarya" and not "higher Eukarya", since "lower Eukarya" is used? Line 181 – orthologs.

Figure 1B – the background color scheme is not defined anywhere. Also, "Euryarchaeota" and "Crenarchaeota" should be used, rather than Euryarchaea and Crenarchaea. Why include an artificial yeast Alg13/14 fusion and not the true sequences? Also, maybe I am getting too old to see tiny figures, but it is really hard to follow the color-based sequence alignment presented. How are the red and yellow residues at the end of the Leishmania sequence conserved since they are only found in that one sequence?

Figure 1C – why not used the same color-coding as used in Figure 1A and 1B? The legend says the Alg13 domain is in turquoise – it is not.

Page 8, line 204 – eukaryotic

Page 8, line 211 – what the RMSD of the homology model? In other words, how good is the fit?

Page 8, line 223 – "both"? E114 and, I assume, H14. Say it better.

Page 8, line 224 – is it known that all the archaeal species assigned to GT28 produce pseudomurein or is this just being assumed?

Page 8, line 226 – "the different acceptor molecule"? As the reader may not be an expert in the field, a reminder of the different acceptor molecules might be in place here.

Page 9, line 233 – a gene (agl24) is essential, not the protein product.

Page 9, line 235 – agl24 not Agl24

Is it possible to create a S. acidocaldarius strain lacking agl24 but in which either the MurG- or Alg13/14-encoding genes are introduced?

Figure S3 – The figure title talks of agl23 deletion. Also, agl24 not Agl24 throughout.

Page 9, line 252 – As written, the sub-heading implies that GlcNAc alone can serve as acceptor. This was not shown.

Figure S4A – Without a control showing the subcellular fraction of another S. acidocaldarius protein of known distribution, the membrane association of Agl24 is not confirmed. Is the Agl24 in the supernatant spill from the membrane lane?

Figure S4C – what does the light blue background represent? What do E and H under the sequence mean (H, hydrophobic; E, ?).

Page 9, lines 258-263 – The reader is directed to Figure 3C, then to Figure 3A and Figure 3B, then to Figure 3D and Figure 3E. Reorder the panels or the text to go from A-E.

Figure 3 – If the in vitro reaction was allowed to run for a prolonged period, was Agl24 able to add an additional GlcNAc?

Page 10, line 274 – reference?

Figure 4 – In each panel, there remain the frames of rectangles used to cover up apparently unrelated values of around the values above each peak. Very sloppy! The last sentence should read "Above 50°C, the increase in the level of the product at the expense of the level of the acceptor is clearly visible".

Page 11, line 292 – Alanine substitution was not inactive. Rather enzymes in which alanine was substituted for histidine were inactive.

Page 11, line 300 – As written, the text leads the reader to believe that only H14 and E114 are conserved. E114 is a second conserved residue, not the second conserved residue.

Figure 5 – The previous section described H14 as being important. Here, H15 is addressed. Also, in each panel, there remain the frames of rectangles around the values above each peak. Again, very sloppy!

Page 14, line 353 – Do the Eps-like sequences in the Euryarchaeota include any of the known enzymes experimentally confirmed as being involved in adding linking sugars?

Page 14, line 363 – "within them" – who is them? Members of the TACK superphylum or in some Euryarchaeota? Not clear.

The entire "The eukaryotic GTs…." section is very difficult to follow. The addition of more narration to this section would make it more readable.

Figure 7 – The use of so much color is distracting. Why does each phylum need a different color? Bootstrap values should be provided to attest to the support given to each clade. How many iterations were run?

Discussion

Page 15, line 403 – 'respectively', not 'either'.

Page 17, line 454 – Does Pyrobaculum calidifontis contain an AglC homologue?

Page 17, line 458 – The authors state: "comparison of archaeal molecular

processes has gradually revealed a strong resemblance to those found in eukaryotes". This is more true of the Crenarchaeota than of the Euryarchaeota.

Page 17, line 463 – "A" current hypothesis.

*Reviewer #2 (Recommendations for the authors):*

I suggest that you slightly change the way you introduce your identification of Alg24 by first describing how you modelled the structure of your candidate enzyme and identified the conserved amino-acids before naming the enzyme and compare Alg24 with its homologues. I suggest that you include EpsF in this comparison. You should put more emphasis on the identification of the motif GGxGGH motif. You could possibly mention and discuss the genetic work in the discussion part. For the phylogenetic analysis, I suggest that you perform a concatenation (removing Methanopyrus kandleri which is a fast-evolving species and disturb the phylogeny with Alg14) and consider my comments about the different subgroups. It should be important having more details about the phylogeny of EpsF and eukaryotes. In fact, your paper is sufficiently interesting and important in the fields of N-glucosylation, that you don't need to oversell it through the fashionable 2D domains story and the Asgard origin of eukaryotes. You repeatedly claim (and illustrate in the cover art) that your phylogenetic analysis indicates that eukaryotic have close homologues in Asgard, suggesting an Asgard origin of these proteins. On the contrary, an interesting aspect of your paper for me is that most Asgard have no Agl13/14 homologues, except a few MAGs that have a distant one. This should be confirmed by the analysis of more Asgard. If the lack of Agl13/14 homologues is confirmed, it will be crystal clear that the few sequences found in Asgard MAGs are not representative of Asgard, in general, and that, even in a 2D perspective, Alg13/14 did not originated from Asgard. You should definitely remove from the title, covert art and so on all misleading sentences with respect to the archaeal and/or Asgard origin of eukaryotes.

*Reviewer #3 (Recommendations for the authors):*

Considering the similarity between N-glycosylation of Eukaryotes and Archaea belonging to TACK superphylum, in the text it should be stressed that GTs from thermophilic Archaea could be excellent models to study the structure/function relationship of elusive eukaryotic enzymes. Their stability could be of great help for detailed characterization and 3D structure. The authors could comment this in the text.

Similarly, the authors should stress that S. acidocaldarius could be used as model system in vivo to proof the function of archaeal homologs of eukaryotic GTs.

Alg24 is homolog to MurG and Alg13/14 belonging, respectively to CAZy families GT28 and GT1, but the low sequence identity left Alg24 unassigned. The scientists in the field of glycobiology would be happy to know in which GT family can be classified Alg24. It would be of great help for others working in this field.

---

## [Author Response]

Essential revisions:1) Revise text for accuracy and readability as noted by the reviewers. Revise title, summary, discussion, and cover art to align with data provided.2) Include statements to make the impact of the work on glycobiology more clear for the reader. Strengthen the messaging of the study as it relates to areas of emphasis noted by the reviewers (use of model system, motif, etc).3) Revise the phylogenetic analysis to strengthen it.4) Figure S4. Provide a control showing the subcellular fraction of another S. acidocaldarius protein of known distribution, to confirm the membrane association of Agl24. If authors are unable to provide this, then the text should be edited appropriately to make the conclusions tentative pending further confirmation in line with evidence provided.

We have conducted all essential revisions.

Reviewer #1 (Recommendations for the authors):General comment – While the actual experiments are solid, the manuscript is very poorly written and even sloppy at places. Too many imprecise statements, errors in figures, etc (see examples, below). The manuscript would also be improved by a quick round of English language editing.My specific comments are listed as encountered while reading the manuscript:IntroductionPage 1, line 26 (Cover Art) – in the upper left-hand corner, an outline of something remains. Remove it.

We thank the reviewer for recognizing this outer line, which appeared after the conversion into the pdf format. Based on the other reviewer comments we have removed the cover art.

Page 2, line 29 (and Page 2, line 54) – In Bacteria, N-glycosylation is seemingly limited to the δ/epsilon proteobacteria. As such, the statement: "Protein N-glycosylation is the most common posttranslational modifications found in all three domains of life" is inaccurate. Indeed, the authors themselves note this point (page 3, line 66).

The sentence has been changed to:

“Protein N-glycosylation is a posttranslational modification found in all three domains of life"

Page 2, line 47 – remove comma between "histidine" and "within".

The sentence has been corrected.

Page 2, line 55 – does eLife require references to be listed alphabetically or chronologically? Here, neither rule applies. Add a period at the end of the sentence.

We have changed the in-text citations to chronological order.

Page 2, line 56 – the text reads: "In Eukarya, N-glycosylation is an essential process", yet on line 61, we are told: "the biological functions of protein glycosylation range from relatively minor to crucial for survival". The two phrases don't match.

We agree with the reviewer that the two sentences might confuse the reader. However, in the first sentence the essentiality is restricted to the N-glycosylation of Eukaryotes, while the second sentence describing the effect protein of glycosylation a broader view (N-,O-, P-, C-glycosylation in all domains of life). We have therefore rewritten the paragraph.

Page 3, line 69 – what about the non-lipid-dependent N-glycosylation that occurs in some bacteria (e.g. Haemophilus influenzae HMW1 adhesin or the actions of NGT, the N-glycosylating GTase in Actinobacillus pleuropneumoniae )?

We thank the reviewer for this reminder. We have added the non-lipid-dependent N-glycosylation of Bacteria and changed the references accordingly.

Page 3, line 74 – to the best of my knowledge, the extended bacterial sequon D/E-X1-N-Χ2-S/T (where X is not proline) is recognized in Campylobacter jejuni and Helicobacter pullorum, the OST of other bacteria show a more relaxed specificity (for instance, see Ollis et al., 2015 Sci Reports).

We had added the more stringent requirement of *Campylobacter jejuni* and *Helicobacter pullorum* OSTs of an acidic residue in the −2 position of the sequon.

“Target proteins contain a specific asparagine residue found within the conserved sequon (N-X-S/T, X ≠ proline), which is recognized by the OST. However, some bacterial OSTs, *e.g.,* those of *Campylobacter jejuni*, require an acidic residue at the −2 position of the sequon (D/E-X_1_-N-Χ2-S/T, X_1_, Χ2 ≠ P) for successful transfer (Ollis *et al.*, 2015).”

However, we later removed this paragraph, as this information is unnecessary to follow the study.

Page 3, line 77 – The statement: "Eukarya require an OST complex of nine subunits" is not accurate. Trypanosomes use a single subunit-containing OST.

The sentence has been expanded accordingly to account for different complexities of the OST in Eukaryotes.

Page 3, lines 78-82 – The writing here is very sloppy. In lines 78-79, we are told: "In contrast, Archaea use the shorter version of the recognition sequon (N-X-S/T) as well as dolichol-phosphate (Dol) as lipid", yet lines 81-82 read: "while the Crenarchaeota use Dol pyrophosphate as in eukaryotes".

The sentence has been rewritten to clarify the differences among the bacterial, eukaryotic, and archaeal N-glycosylation.

Page 4, line 103 – "This is a rare example of cross-domain complementation" is inaccurate. There are numerous examples of complementation of a eukaryote mutant by a bacterial gene (for instance, see Kachroo et al., eLife 2017 for examples). Moreover, no hypothesis of a common origin for eukaryotic and archaeal N-glycosylation was stated up to this point, so it is not clear how this statement further strengthens such a concept.

The sentence has been deleted, as this statement is unnecessary.

Page 4, lines 108-109 – Agl24 would not generate the lipid-linked chitobiose core; rather, it would contribute to the assembly of this moiety.

We have now changed the sentence to:

“Activity assays, along with HPLC, MALDI, and NMR analyses confirmed the function of Agl24 as catalyzing the second N-glycan biosynthesis step to generate the lipid-linked chitobiose core.”

Page 4, line 110 – On lines 103-104, the reader is told "of a common origin of eukaryotic and archaeal N-glycosylation", yet here it is proposed that "the eukaryotic N-glycosylation pathway originates from an archaeal ancestor". These scenarios are not really the same thing.

We thank the reviewer for noticing this. We meant that the eukaryotic and archaeal N-glycosylation share a common ancestor (which they technically do, even if the eukaryotic homologs branch from within Archaea), the phrase is not appropriate. We have kept the sentence “Our extensive bioinformatics analyses support the hypothesis that the eukaryotic N-glycosylation pathway originates from an archaeal ancestor.”, as it is in our opinion the optimal way to phrase this.

Larkin et al., (2013, Nat Chem Biol), where AglK was shown to add GlcNAc to dolichol phosphate in the methanoarchaea Methanococcus voltae should also be addressed in the Introduction, especially given how genetic studies on this organism have assigned this role to AglH (Shams-Eldin et al., 2008, J. Bacteriol.).

We agree with the reviewer and have added the following text:

“This functional complementation has been first described for the AglH of Methanococcus voltae (Shams-Eldin et al., 2008), which has suggested that AglH initiates the N-glycosylation by transferring GlcNAc-1-P onto Dol-P. However, a biochemical study clearly showed that AglK is responsible for the generation of Dol-P-GlcNAc, on which the N-glycan is assembled (Larkin et al., 2013). This observation agrees that Euryarchaeota build their N-glycans on Dol-P and not on Dol-pyrophosphate in a similar manner to Crenarchaeaota and Eukaryotes (Taguchi et al., 2016).”

Likewise, the N-linked glycan of the crenarchaeote Pyrobaculum calidifontis, which includes a core reminiscent of chitobiose (Fujinami et al., 2017, Glycobiology) should also be noted in the Introduction.

We agree and have added following text:

“Recently, the characterization of the N-glycan from Pyrobaculum calidifontis, belonging in the Thermoproteales order of the Crenarchaeota, revealed as similar N-glycan chitobiose core structure, with the difference that both GlcNAc are modified with an additional N-acetyl group at the C3 position (Fujinami et al., 2017).”

ResultsPage 5, line 115 – "this"?

We have changed “this” to: “the”

Page 5, lines 123-124 – Not accurate. CAZy lists the halophile Salinadaptatus halalkaliphilus, a member of the family Natrialbaceae, as a GT28 family member (http://www.cazy.org/GT28_archaea.html). Also, Kandler and/or Konig should be cited for their work on pseudomurein.

We thank the reviewer for pointing this out. We have added to the text that

a GT belonging to the halophilic *Salinadaptatus halalkaliphilus* was recently grouped with the GT family 28 in the CAZy database;

We have also amended the citations to refer to the work of Kandler and Konig for the analyses and description of pseudomurein (Kandler and König, 1978, König et al., 1982).

Page 5, line 142 – The authors should refer to the paper by Cavalier-Smith and Chao (2020, Protoplama) where Saci1262 is described as corresponding to MurG. Indeed, this reference addresses similarities between MurG, Alg13/14 and Saci1262.

We have added the following text:

“In addition, phylogenetic studies indicate similarities of MurG and Alg14-13 with Saci1262 (Cavalier-Smith and Chao, 2020, Lombard, 2016).”

Page 6, line 148 – At this stage of the story, there is no justification in assigning Saci1262 to the N-glycosylation pathway, namely, assigning it the Agl tag used for confirmed archaeal N-glycosylation pathway components. Such involvement needs to be first demonstrated – no such demonstration has been presented at this stage of the report.

We agree with the reviewer. We now assign Saci1262 as archaeal glycosylation enzyme 24 (Agl24) after we have demonstrated its function.

Page 6, lines 149-153 – Since the reader is not told where other N-glycosylation genes are found relative to Saci1262, this bit of text could confuse those outside the field. Indeed, the authors finish the section by stating how Saci1262 is only eight genes downstream of the OST-encoding aglB gene, leaving the reader unclear as to how N-glycosylation gene clustering is defined.

We have added the following sentence to the text:

“Interestingly, the gene *saci1262* is located only eight genes downstream of the OST aglB (*saci1274*) in S. acidocaldarius (Figure S1), while the genes coding for the identified enzymes involved in the N-glycosylation, e.g. *aglH (saci0093*), *agl3 (saci423*), or *agl16 (saci0807*), are found distantly far separated scattered over across the entire genome.”

The "Detailed bioinformatics….." section beginning on line 156 would be easier to read if the authors better separated data based on sequence comparisons from data based on structural comparisons. Moreover, including evolutionary considerations in the middle of the first paragraph of the section (lines 163-168) further adds to mixture of ideas that require better organization.

The whole section has been rearranged to separate the data based on the sequence and data based on the structural comparison. Additionally, the sentence on the evolutionary considerations has been removed.

Page 6, line 170 – "archaeal Agl24 enzymes"? At this point, the reader has only been told of Saci1262. Figure 1B and Figure S2, where archaeal Saci1262 homologues are listed, have not yet been cited in the text. Indeed, without knowing anything about N-glycosylation in the majority of these species, it is premature to assign them the Agl24 title – at this stage they are homologues of Saci1262.

We agree; we now assign Saci1262 as archaeal glycosylation enzyme 24 (Agl24) after we have demonstrated its function (see above).

Figure 1A – "Archaea" should not be italicized. Also, why "most Eukarya" and not "higher Eukarya", since "lower Eukarya" is used? Line 181 – orthologs.

Thank you for pointing this out. In addition, we have switched all mentions of “higher” and “lower” Eukaryotes to “recently-” and “deep-branching” to be more correct from an evolutionary standpoint.

Figure 1B – the background color scheme is not defined anywhere. Also, "Euryarchaeota" and "Crenarchaeota" should be used, rather than Euryarchaea and Crenarchaea. Why include an artificial yeast Alg13/14 fusion and not the true sequences? Also, maybe I am getting too old to see tiny figures, but it is really hard to follow the color-based sequence alignment presented. How are the red and yellow residues at the end of the Leishmania sequence conserved since they are only found in that one sequence?

A description of the background color has been added to the legend. The terms Euryarchaeota and Crenarchaeota are now used. We used an artificial fusion of Alg14-13 as the genes are split in yeast, to enable the comparison. Indeed, in the alignments the Weblogo highlights at the end of the sequence of Leishmania a lysine (K) residue. This conservation does not make sense at it is the only sequence extending so far, thus it was corrected. A larger alignment is now shown instead in Figure S2.

Figure 1C – why not used the same color-coding as used in Figure 1A and 1B? The legend says the Alg13 domain is in turquoise – it is not.

We now use the same color-coding for Figure 1A and 1B (Bacteria: light and dark violet, Eukarya: light and dark gray, and Crenarchaeota: light and dark orange)

Page 8, line 204 – eukaryotic

This has been corrected.

Page 8, line 211 – what the RMSD of the homology model? In other words, how good is the fit?

To give a better estimation of the fit of our structural homology model we added the Root Mean Square Deviation (*RMSD*) values in Table S2. Also, in table S2 the Swiss Model fits are displayed, e.g., the Global Model Quality Estimation (GMQE), quaternary structure quality estimate (QSQE), and sequence identity to the different known Structures of MurG and Alg13.

We also ran ALPHAFOLD, which predicted a similar model we had obtain from Swiss model, we also include this in the text and table S2 (RMSD ALPHAFOLD vs Agl24 model: 3.80).

Page 8, line 223 – "both"? E114 and, I assume, H14. Say it better.

We have now specified E114 and H14.

Page 8, line 224 – is it known that all the archaeal species assigned to GT28 produce pseudomurein or is this just being assumed?

All Archaea possessing a GT, which are grouped to the Family GT28, are known to produce a pseduomurein cell wall. This includes the members of the archaeal order Methanobacteriales (e.g. *Methanothermus fervidus*, *Methanosphaera stadtmanae*, and *Methanobrevibacter formicicum*) and the order Methanopyrales (*Methanopyrus kandleri*), with the exception of the newly discovered *Salinadaptatus halalkaliphilus* 2447, for which no information of the cell-wall composition exists. We therefore have added the following text:

“of known pseudomurein-producing Archaea of the order Methanobacteriales and Methanopyrales.”

Page 8, line 226 – "the different acceptor molecule"? As the reader may not be an expert in the field, a reminder of the different acceptor molecules might be in place here.

Thank you for bringing this to our attention. We have added the following:

“We propose that these residues might be required to accommodate the different sugar N-acetyltalosaminuronic acid of the lipid-linked acceptor molecule in the pseudomurein biosynthesis process.”

Currently, there is only the proposed biosynthesis process of pseudomurein reported by Kandler and König (1993) “Cell envelopes of archaea: Structure and chemistry”, which proposed a similar biosynthetic pathway for pseudomurein. However, at that time it was predicted that UDP-MurNAc-peptides and UDP-GlcNAc are linked prior the transfer of this di-saccharide-peptide-linker to the lipid carrier.

Page 9, line 233 – a gene (agl24) is essential, not the protein product.

This has been corrected.

Page 9, line 235 – agl24 not Agl24

This has been corrected.

Is it possible to create a S. acidocaldarius strain lacking agl24 but in which either the MurG- or Alg13/14-encoding genes are introduced?

Indeed, it is possible to integrate genes in the genome of *S. acidocaldarius*. This was done for the oligosaccharyltransferase gene *aglB* to show that the original *aglB* can be deleted, when a second copy is present in the genome at a different location (Meyer and Albers 2014):

“AglB catalyzing the oligosaccharyl transferase step of the archaeal N-glycosylation process is essential in the thermoacidophilic crenarchaeon *Sulfolobus acidocaldarius”.*

However, based on the thermophilic growth condition of *S. acidocaldarius* at 75°C a complementation with a eukaryotic Alg14-13 is not possible. There are, of course, thermostable homologs of the bacterial MurG, *e.g*., of *Hydrogenivirga*, which could be used for an integration. However, as the N-glycosylation is essential in *Sulfolobus acidocaldarius* and the transposon mutagenesis study in *S. islandicus* confirmed our finding of the essential properties of *saci1262* (*agl24*), we did not proceed with a complementation approach.”

Figure S3 – The figure title talks of agl23 deletion. Also, agl24 not Agl24 throughout.

We thank the reviewer for pointing this out. We have corrected this mistake. During the writing of the manuscript the genes *agl22* and *agl23* in *Haloarcula hispanica* were described, therefore the numbering was changed to *agl2*4 for *saci1262.* The text has been corrected accordingly.

Page 9, line 252 – As written, the sub-heading implies that GlcNAc alone can serve as acceptor. This was not shown.

The sub-header has been changed to:

“Agl24 transfers a single GlcNAc residue onto a GlcNAc pyrophosphate linked acceptor molecule” to avoid any confusion.”

Figure S4A – Without a control showing the subcellular fraction of another S. acidocaldarius protein of known distribution, the membrane association of Agl24 is not confirmed. Is the Agl24 in the supernatant spill from the membrane lane?

We agree with the reviewer. We have conducted additional experiments. Saci1262(Agl24) was purified together with the soluble protein Agl3 (UPD-Sulfoquinovose synthase) to show that the cellular localization is distinct from Agl3. However, due to the lack of a “Duet” expression plasmid for *S. acidocaldarius*, two strains expressing either the Saci1262(Agl24) or Agl3 were mixed (1:1 ratio), followed by the separation of the membrane and soluble fraction by ultracentrifugation. These new results revealed that Agl3 is only found in the soluble fraction, while Saci1262(Agl24) is also found in the membrane faction (Figure S4 A and B).

Figure S4C – what does the light blue background represent? What do E and H under the sequence mean (H, hydrophobic; E, ?).

The light blue background was removed and the following text has been added to the legend:

“Secondary protein structure prediction by the Jpred Server. Top line corresponds to the amino acid sequence of Saci1262/Agl24, the second line represent extended (E), helical (H) and other (-) types of secondary structure, respectively.”

Page 9, lines 258-263 – The reader is directed to Figure 3C, then to Figure 3A and Figure 3B, then to Figure 3D and Figure 3E. Reorder the panels or the text to go from A-E.

We have changed the text starting with Figure 3A and C. However, to enable a better comparison we have retained the order of the panels to show the substrates’ MALDI-MS spectra directly above the product spectrum of each acceptor molecule.

Figure 3 – If the in vitro reaction was allowed to run for a prolonged period, was Agl24 able to add an additional GlcNAc?

We were not able to see a peak corresponding to the mass of an additional GlcNAc (1033 *m/z* [M-2H^+^3Na]^+^ Acceptor-1 or 1100 m/z [M-2H^+^3Na]^+^ Acceptor-2). Therefore, we have added the following sentence: “With an extended reaction time, no further additions of GlcNAc by Agl24 could be detected.” SF6 A shows a representative HPLC trace of the enzymatic reaction, revealing that no additional product peaks are formed in the Saci1262(Agl24) assay.

Page 10, line 274 – reference?

The reference “Brock *et al.,* 1972” has been added.

Figure 4 – In each panel, there remain the frames of rectangles used to cover up apparently unrelated values of around the values above each peak. Very sloppy! The last sentence should read "Above 50°C, the increase in the level of the product at the expense of the level of the acceptor is clearly visible".

During the conversion to a pdf file, the empty text boxes of negligible peaks became visible. They have now been deleted. The last sentence has been modified based on the reviewer’s suggestion.

Page 11, line 292 – Alanine substitution was not inactive. Rather enzymes in which alanine was substituted for histidine were inactive.

The text has been corrected.

Page 11, line 300 – As written, the text leads the reader to believe that only H14 and E114 are conserved. E114 is a second conserved residue, not the second conserved residue.

The word “second” has been removed.

Figure 5 – The previous section described H14 as being important. Here, H15 is addressed. Also, in each panel, there remain the frames of rectangles around the values above each peak. Again, very sloppy!

The text and visible frames have now been corrected.

Page 14, line 353 – Do the Eps-like sequences in the Euryarchaeota include any of the known enzymes experimentally confirmed as being involved in adding linking sugars?

To our knowledge, none of these Eps-like homologs in Euryarchaeota has been characterized.

Page 14, line 363 – "within them" – who is them? Members of the TACK superphylum or in some Euryarchaeota? Not clear.

The entire section on the evolution of Agl24 and its homologs has been rewritten, in light of having used an expanded set of genomes for our analyses. Originally, the sentence was referring to the various archaeal clades, i.e., there existing an Agl24 clade among them. We do concur that the statement was ambiguous.

The entire "The eukaryotic GTs…." section is very difficult to follow. The addition of more narration to this section would make it more readable.

We would like to thank the reviewer for this comment and we hope that with this revised section, it will be more straightforward to follow. Some details on the evolution of these homologs have been added to the manuscript. Nonetheless, we would like to point out that the purpose was to focus more on Agl24 itself and the relationship of the eukaryotic and Asgard sequences with one another and the remaining homologs, instead of every single archaeal homolog. In addition, to go into more detail, we added the uncollapsed phylogenies in the Supplementary Information and Supplementary Data 1 and refer to these, to avoid cluttering the Results sections.

Figure 7 – The use of so much color is distracting. Why does each phylum need a different color? Bootstrap values should be provided to attest to the support given to each clade. How many iterations were run?

While the plan initially was to use separate colors to help make each lineage distinct, we have heeded the reviewer’s advice and only added color to the most important clades (Agl24, Eukaryotes, Asgards), as described in the text. Furthermore, we have removed the background color, since we agree that it was distracting. As described in the Figure 7 legend, the black dots denote strongly supported clades (ultrafast boostrap >=95, aLRT SH-like >=80), as suggested in the IQ-TREE documentation and the supports are also provided in the uncollapsed phylogenies now provided as Supplementary Figures. The number of replicates (1000 each for ultrafast bootstraps and aLRT SH-like) is mentioned in the corresponding part of the Methods section.

DiscussionPage 15, line 403 – 'respectively', not 'either'.

This has been corrected.

Page 17, line 454 – Does Pyrobaculum calidifontis contain an AglC homologue?

There exists a potential AglC homolog in *P. calidifontis*. We have added the following:

“The best candidate for an AglC homolog in *P. calidifontis* is Pcal_0481 showing 29.91% sequence identity. Interestingly, the structure has recently been described, while the predicted function as a mannosyltransferase could not be experimentally confirmed (Gandini et al., 2020).”

Page 17, line 458 – The authors state: "comparison of archaeal molecularprocesses has gradually revealed a strong resemblance to those found in eukaryotes". This is more true of the Crenarchaeota than of the Euryarchaeota.

The sentence has been revised to:

“Since the discovery of the Archaea as the third domain of life along with Bacteria and Eukaryotes (Woese and Fox, 1977), the comparison of archaeal molecular processes, especially those found in Crenarchaeota, has gradually revealed a strong resemblance the ones found in Eukaryotes (Huet et al., 1983, Zillig et al., 1989, Akil and Robinson, 2018, Akil et al., 2020).”

Page 17, line 463 – "A" current hypothesis.

The sentence has been changed.

Reviewer #2 (Recommendations for the authors):I suggest that you slightly change the way you introduce your identification of Alg24 by first describing how you modelled the structure of your candidate enzyme and identified the conserved amino-acids before naming the enzyme and compare Alg24 with its homologues. I suggest that you include EpsF in this comparison. You should put more emphasis on the identification of the motif GGxGGH motif.

We thank the reviewer for this suggestion. Based on the lack of structural information on EpsE/F enzymes, they were not included, only the well described MurG and Alg14-13.

You could possibly mention and discuss the genetic work in the discussion part.

We added the information that the essentiality is a good indication that this enzyme is the most likely candidate to conduct the second N-glycosylation step.

For the phylogenetic analysis, I suggest that you perform a concatenation (removing Methanopyrus kandleri which is a fast-evolving species and disturb the phylogeny with Alg14) and consider my comments about the different subgroups. It should be important having more details about the phylogeny of EpsF and eukaryotes.

As we explained above, we did run a concatenated phylogeny of the two proteins although it is not good practice in this case, due to the incongruences between the individual phylogenies. Additional details about the EpsEF and eukaryotic clades can be gleaned from the Supplementary Figures, but they were not the focus of this manuscript and we did not use the full range of sequences in the final phylogenies (mainly due to the computational load and suspecting that divergent sequences would reduce the quality of the alignment).

In fact, your paper is sufficiently interesting and important in the fields of N-glucosylation, that you don't need to oversell it through the fashionable 2D domains story and the Asgard origin of eukaryotes. You repeatedly claim (and illustrate in the cover art) that your phylogenetic analysis indicates that eukaryotic have close homologues in Asgard, suggesting an Asgard origin of these proteins. On the contrary, an interesting aspect of your paper for me is that most Asgard have no Agl13/14 homologues, except a few MAGs that have a distant one. This should be confirmed by the analysis of more Asgard. If the lack of Agl13/14 homologues is confirmed, it will be crystal clear that the few sequences found in Asgard MAGs are not representative of Asgard, in general, and that, even in a 2D perspective, Alg13/14 did not originated from Asgard. You should definitely remove from the title, covert art and so on all misleading sentences with respect to the archaeal and/or Asgard origin of eukaryotes.

We are inclined to agree with many of these suggestions. The situation of the Asgards regarding N-glycosylation appears to be far more complicated than what would be expected from simply adhering to a 2D scenario. On the other hand, the eukaryotic homologs are of archaeal origin and their closest currently known relatives are Asgards. Therefore, in the revised manuscript we have explored the different implications of our phylogenies. We have removed the cover art to avoid any confusions.

Reviewer #3 (Recommendations for the authors):Considering the similarity between N-glycosylation of Eukaryotes and Archaea belonging to TACK superphylum, in the text it should be stressed that GTs from thermophilic Archaea could be excellent models to study the structure/function relationship of elusive eukaryotic enzymes. Their stability could be of great help for detailed characterization and 3D structure. The authors could comment this in the text.Similarly, the authors should stress that S. acidocaldarius could be used as model system in vivo to proof the function of archaeal homologs of eukaryotic GTs.

We have highlighted the excellent opportunity to study the homologs of the eukaryotic GT in Archaea, and in particular in *S. acidocaldarius*, in the last paragraph.

Alg24 is homolog to MurG and Alg13/14 belonging, respectively to CAZy families GT28 and GT1, but the low sequence identity left Alg24 unassigned. The scientists in the field of glycobiology would be happy to know in which GT family can be classified Alg24. It would be of great help for others working in this field.

We have contacted Prof. Bernard Henrissat, the creator of the CAZy database. Agl24 will be assigned to a new GT family as soon as the paper is accepted and in the proof stage.